# BENCHMARKING ALGORITHMS FOR FEDERATED DOMAIN GENERALIZATION

## ABSTRACT

In this paper, we present a unified platform to study domain generalization in the federated learning (FL) context and conduct extensive empirical evaluations of the current state-of-the-art domain generalization algorithms adapted to FL. In particular, we perform a fair comparison of 11 existing algorithms in solving domain generalization either centralized domain generalization algorithms adapted to the FL context or existing FL domain generalization algorithms to comprehensively explore the challenges introduced by FL. These challenges include statistical heterogeneity among clients, the number of clients, the number of communication rounds, etc. The evaluations are conducted on five diverse datasets including PACS (image dataset covering photo, sketch, cartoon, and painting domains), FEMNIST (image dataset containing writing digits and characters written by more than 3500 users), iWildCam (image dataset with 323 domains), Py150 (natural language processing dataset with 8421 domains) and CivilComments (natural language processing dataset with 16 domains). The experiments show that the challenges brought by federated learning stay unsolved in realistic experimental settings. Furthermore, the code base supports fair and reproducible evaluation of new algorithms with little implementation overhead.

## 1 INTRODUCTION

Federated learning (FL) Konečnỳ et al. (2016) is a distributed machine learning approach that assumes each client or device owns a local dataset and this local dataset cannot be exchanged or centrally collected because of privacy or communication constraints. Given this context, a natural paradigm for FL (e.g., FedAvg McMahan et al. (2017)) is to alternate between two stages: clients locally update the model based on its local dataset and a central server aggregates client models. Because the clients may be phones, network sensors, hospitals, or alternative local information sources, the local datasets are naturally heterogeneous between clients. Specifically, there are at least two types of realistic statistical data heterogeneity in the FL context. *Client heterogeneity* is the data heterogeneity between clients involved in training—e.g., hospitals may use different staining procedures or imaging equipment. *Train-test heterogeneity* is the data heterogeneity between the training and testing data—e.g., the performance on a new client that was not involved in training or a natural shift in real-world test data due to changes over time, location, or context.

Client heterogeneity has long been considered a statistical challenge since federated learning was introduced. FedAvg McMahan et al. (2017) has experimentally shown that their methods effectively mitigate some client heterogeneity. There are many other extensions based on the FedAvg framework tackling the heterogeneity among clients in FL Hsieh et al. (2020); Li et al. (2020); Karimireddy et al. (2020). There is an alternative setup in FL, known as the personalized setting, which aims to learn personalized models for different clients to tackle heterogeneity. Numerous recent papers have proposed FL models and algorithms to accommodate personalization Smith et al. (2017); Chen et al. (2018); Hanzely et al. (2020); T Dinh et al. (2020); Deng et al. (2020); Acar et al. (2021). However, these prior works only train the model on simple data sets such as MNIST, EMNIST, and CIFAR10 and the client heterogeneity is constructed mainly through *class imbalance*, which assumes the ratio of data from each class is different for each client but the class conditional distributions are homogeneous across clients. Class imbalance is a special kind of heterogeneity called prior probability shift. In practice, due to the difference between the location of the local data collector (cameras, sensors, etc), real data heterogeneity is more complex than simple class

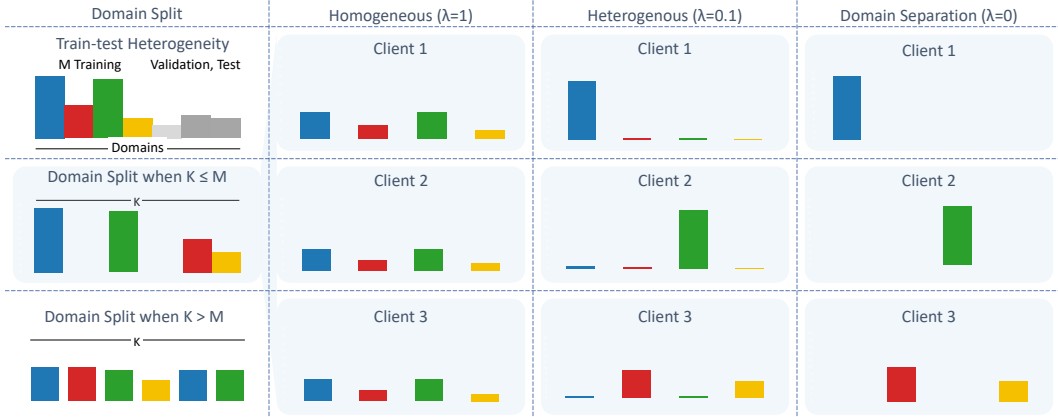

Figure 1: Our benchmark simultaneously evaluates *train-test heterogeneity* (i.e., domain generalization) as seen in the top left and *client heterogeneity* across domains by splitting the domain data amongst clients. The client heterogeneity can be homogeneous (left), heterogeneous (center), or domain separated (right), where $M$ is the number of domains, $K$ is the number of clients, color denotes domain data, and $\lambda$ is the domain balance parameter. The right three panels demonstrate the domain split for homogeneous, heterogeneous, and domain separation when $K \leq M$.

imbalance as in many prior works. Furthermore, most prior FL works do not consider train-test heterogeneity but rather assume the train and test data are i.i.d.

While most FL works have usually ignored train-test heterogeneity, the task of *domain generalization* (DG) Blanchard et al. (2011) formalizes a special case of train-test heterogeneity in which the training algorithm has access to data from multiple source domains and the goal is to perform well on data from an *unseen* test domain. There is an active line of research on domain generalization in the centralized setting Muandet et al. (2013); Saito et al. (2018); Ganin & Lempitsky (2015); Long et al. (2015); Arjovsky et al. (2019); Sagawa et al. (2019); Shi et al. (2021); Li et al. (2018). Current benchmark papers Koh et al. (2021); Gulrajani & Lopez-Paz (2020) provide thorough comparisons between different algorithms for DG. However, these centralized DG benchmarks do not consider the unique constraints of the FL context. In particular, they fail to provide insights on how the client dataset heterogeneity, the number of clients, and the communication budget will influence the generalization ability. From the FL side, to the best of our knowledge, there are currently only four published works, FedADG Zhang et al. (2021), FedDG Liu et al. (2021), FedSR Nguyen et al. (2022) and FedGMA Tenison et al. (2022) that tackle DG in the FL context but their evaluations are limited in the follwing senses: **1)** The evaluation datasets are limited in the number and diversity of domains. FedDG Liu et al. (2021) only evaluates on retinal fundus images with 4 domains and prostate T2-weighted MRI images from 6 domains, and FedADG only focuses on PACS and VLCS, which share similarities in terms of number of domains (each with only 4 domains) and sample size. FedGMA Tenison et al. (2022) only focuses on ColorMNIST with a few domains. However, we consider more realistic datasets with thousands of domains. See Sec. 4.2 for details. **2)** Their evaluations are restricted to the case when the number of clients is equal to the number of domains, which may be an unrealistic assumption (e.g., a hospital that has multiple imaging centers or a device that is used in multiple locations). For example, FedSR Nguyen et al. (2022) and FedADG Zhang et al. (2021) only evaluates on the case when the number of clients equals to the number of domains. However, we show in this paper, FedSR Nguyen et al. (2022) and FedADG Zhang et al. (2021) are sensitive to clients number, and they fail even on the simple dataset PACS when the clients number large than 20. See Sec. 4.3 massive number of clients for detail. **3)** They fail to show the network effect of FL; in particular, neither work considers the influence of the number of clients on the performance, and FedADG Zhang et al. (2021) does not consider the effect of the number of communication rounds. In summary, the current DG benchmarks fail to consider challenges unique to FL, and the few FL methods for DG have limited evaluations.

Therefore, more systematic evaluation is needed both to aid in systematic progress at the intersection of FL and DG but also to aid in answering many open questions. For example, how does client

domain heterogeneity influence the performance of current algorithms? What is the performance of a direct translation of centralized DG algorithms (if applicable) to the FL context? How does the performance of current algorithms scale with the number of clients and communications rounds on complicated real-world datasets?

**Major contributions:** This work addresses the above questions, and our contributions can be summarized as follows. **1)** We propose a standardized definition of *client domain heterogeneity* that is unique to the FL context and interpolates between domain homogeneity and domain separation (see Figure 1) while limiting the class imbalance. In particular, we develop an experimental setup method to split dataset domain samples among any number of clients (see subsection 3.1). **2)** We provide a fair comparison over multiple representative centralized DG methods adapted to the FL context as well as four prior works on federated domain generalization on five different benchmark datasets. **3)** We also explore the impact on the generalization ability of client domain heterogeneity, the total client number, communication rounds, which are unique to the FL context. **4)** From these results, we identify significant generalization gaps between centralized domain generalization and domain generalization in the federated Learning setting. **5)** We release an extensible open-source code library for studying domain generalization in the FL context (see https://github.com/anonymous-lab-ml/benchmarking-dg-fed).

## 2 BACKGROUND AND RELATED WORK

**Domain generalization.** Domain generalization (DG) task Blanchard et al. (2011) tackles the classification problem where the domains of test and training dataset are heterogeneous, i.e., the test data comes from an unseen domain distribution. Existing DG algorithms design alternative objectives seeking for a good approximation usually by utilizing training data from multiple domains. Schölkopf et al. (2021) claims that current machine learning model only captures superficial statistical correlation rather than underlying causal relationship between data and their labels. Inspired by causality Lopez-Paz et al. (2017); Peters et al. (2016); Heinze-Deml et al. (2018), Arjovsky et al. (2019) proposes IRM seeking invariant predictor on top of the representation. Li et al. (2018) instead, explicitly tries to align the feature representation distribution by minimizing the maximum mean discrepancy (MMD) Gretton et al. (2006). Similarly, Sun & Saenko (2016) proposes CORAL which utilizes the second-order information. Besides invariant representation, Shi et al. (2021) solves the DG problem using regularization, the proposed FISH algorithm is a first-order method aligning the gradient directions generated by data from different domains. Sagawa et al. (2019) introduces the objective that tackles the worst-performed domain. The proposed algorithm did not directly optimize the objective by solving the minimax objective, but to solve ERM while increasing the importance of domains with larger loss during training. Zhang et al. (2017) proposes a straightforward data augmentation principle called Mixup that trains a neural network on convex combinations of data from two distinct domains.

**Federated Domain Generalization:** Limited works seek to solve the DG task in the FL context. Liu et al. (2021) recently proposed FedDG, a federated learning paradigm specifically designed for medical image classification. The proposed method requires sharing the amplitude spectrum of images among local clients. Zhang et al. (2021) applies generative adversarial network (GAN) Goodfellow et al. (2020) in the FL context, where each client contains four models: a featurizer $\Phi$, a classifier $w$, a generator $\mathcal{G}$ and a discriminator $\mathcal{F}$. The featurizer $\Phi$ and classifier $w$ solves the classification task, while the generator $\mathcal{G}$ learns the feature distribution $Z$. During local training, FedADG Zhang et al. (2021)first trains the featurizer $\Phi$ and classifier $w$ by minimizing the empirical loss, then trains the generator $\mathcal{G}$ and the discriminator $\mathcal{F}$ using the GAN Goodfellow et al. (2020) approach. After the local training, the central server aggregates the local featurizer, classifier and generator. However, this method requires training 4 models together along with tens of hyperparameters to tune, which makes it hard to converge empirically. Tenison et al. (2022) proposes a new aggregation method called Federated Gradient Masking Averaging (FedGMA) with the goal of improving generalization across clients and of the global model. Their gradient masking prioritizes gradient components that are aligned with the overall dominant direction across clients while the inconsistent components of the gradient are given less importance. Yuan et al. (2021) introduced the concept participation gap to identify dataset heterogeneity. They train models using a set of participating clients and examine their performance on held-out data from these clients as well as an additional set of non-participating clients. Therefore it is closely related to domain adaptation in FL context, which

is different than the domain generalization setting considered in this benchmark, i.e., the training and test domains do not overlap. Nguyen et al. (2022) proposed FedSR where they enable domain generalization while still respecting the decentralized and privacy-preserving natures of FL context by enforcing $\ell_2$-norm and a conditional mutual information regularizer on the representation. In Table 1, we give an overview of domain heterogeneity across two dimensions: 1) between training and testing datasets (i.e., standard vs domain generalization task) and 2) among clients (i.e., domain imbalance between clients). While some work has considered the standard supervised learning task (left column), a new fair evaluation is needed to understand the behaviour of domain generalization algorithms in the federated context including the influence of data heterogeneity, communication budget, and the number of clients.

Table 1: Different Tasks with Domain Heterogeneity in the FL Context

| Among clients | Between training and testing datasets | |
|---|---|---|
| | Standard Supervised Learning | Domain Generalization (our focus) |
| Homogeneous ($\lambda = 1$) | Standard FL with Domain Homogeneity | Federated DG with Domain Homogeneity |
| Heterogeneous ($0 < \lambda < 1$) | FL with Domain Heterogeneity | Federated DG with Domain Heterogeneity |
| Domain separation ($\lambda = 0$) | FL with Domain Separation | Federated DG with Domain Separation |

## 3  PROBLEM SETUP

### 3.1  DOMAIN HETEROGENEITY.

**Implementation of domain heterogeneity**  We now provide a concrete procedure for implementing domain heterogeneity for the benchmark (see Figure 1 for an illustration). Given the number of training samples for all $M$ domains, denoted by $\{n_m\}_{m=1}^M$ where $n_m$ is the number of samples for domain $m$, we first assign "primary" domains $P_k \subseteq \{1, 2, \ldots, M\}$ to each client via the domain split function defined in Algorithm 1, i.e., $\{P_k\}_{k=1}^K = \text{DomainSplit}(K, \{n_m\}_{m=1}^M)$. Algorithm 1 carefully handles two cases: fewer clients than domains ($K \leq M$) and more clients than domains ($K > M$). In the first case, the domains are sorted in descending order and are iteratively assigned to the client $k^*$ which has the smallest number of training samples $\sum_{m' \in P_{k^*}} n_{m'}$ currently. In this way, the algorithm outputs $\{P_k\}_{k=1}^K$ such that no client shares domains with the others but attempts to balance the total number of training samples between clients. In the case $K > M$, we first assign the domains one by one to the first $M$ clients. Then, starting from client

---

**Algorithm 1** DomainSplit function where w.l.o.g. the domains are assumed to be in descending order, i.e., $n_1 \geq n_2 \geq \cdots \geq n_M$.

**Input** $K, n_1, \ldots, n_M$
  **if** $K \leq M$ **then**
    $\forall k, P_k \leftarrow \emptyset$
    **for** $m = 1, 2, \ldots, M$ **do**
      $k^* \in \arg\min_k \sum_{m' \in P_k} n_{m'}$
      $P_{k^*} \leftarrow P_{k^*} \cup \{m\}$
    **end for**
  **else if** $K > M$ **then**
    $\forall k \in \{1, 2, \ldots, M\}, P_k \leftarrow \{k\}$
    **for** $k = M + 1, \ldots, K$ **do**
      $m^* \in \arg\max_m \frac{n_m}{\sum_{k'=1}^K \mathbb{1}[m \in P_{k'}]}$
      $P_k \leftarrow \{m^*\}$
    **end for**
  **end if**
**Output** $P_k$

---

$k = M + 1$, we iteratively split the largest domain $m^*$, where the samples are evenly split among all clients where $m^* \in P_k$. In this way, some clients may share one domain, but no client holds two domains simultaneously. Again, this also attempts to balance the number of samples across clients as much as possible.

After selecting the primary domains $P_k$ for each client, we define the training sample counts, denoted $n_{m,k}(\lambda)$, for domain $m$, client $k$, and domain balance parameter $\lambda \in [0, 1]$ :

$$n_{m,k}(\lambda) = \lambda \frac{n_m}{K} + (1 - \lambda) \frac{\mathbb{1}[m \in P_k] \cdot n_m}{\sum_{k'=1}^K \mathbb{1}[m \in P_k]} , \tag{1}$$

where rounding to integers is carefully handled when not perfectly divisible and where $\mathbb{1}[\cdot]$ is an indicator function. This is simply a convex combination between a uniform splitting of domains among clients (i.e., the $\frac{n_m}{K}$ term) and a splitting where each client has a disjoint set of domains (i.e., the $\frac{\mathbb{1}[m \in P_k] \cdot n_m}{\sum_{k'=1}^K \mathbb{1}[m \in P_k]}$ term)—unless $K > M$ and then we try to split domains evenly based on number of samples as defined in Algorithm 1. After defining $n_{m,k}(\lambda)$, we can denote the total training samples related to client $k$ with domain balance parameter $\lambda$ as $n_k(\lambda) = \sum_{m=1}^M n_{m,k}(\lambda)$.

## 3.2 FEDERATED DOMAIN GENERALIZATION

In the federated domain generalization problem, we are interested in collaboratively training a model across the clients to perform well on unseen test domains $\mathcal{D} \in \mathcal{D}^{\text{test}}$, which is different from training domains $1, \ldots, M$. Therefore, we focus on minimizing not the average loss on the source domains constructed from each client $f_k(\theta) = \mathbb{E}_{(x,y)\sim\mathcal{D}_k}[\ell(((x,y);\theta))] \approx (1/n_k)\sum_{i=1}^{n_k} \ell(x_k^i, y_k^i; \theta)$, but on the unseen test domains, in either an average (or worst-case sense) as defined below

$$\min_\theta \mathbb{E}_{\mathcal{D}\sim\mathcal{D}^{\text{test}}} \mathbb{E}_{(x,y)\sim\mathcal{D}}[\ell(((x,y);\theta))], \quad \text{or} \quad \min_\theta \sup_{\mathcal{D}\sim\mathcal{D}^{\text{test}}} \mathbb{E}_{(x,y)\sim\mathcal{D}}[\ell(((x,y);\theta))]. \tag{2}$$

To solve equation 2 in the federated learning context, we need to consider client domain heterogeneity, as it can introduce unique challenges for domain generalization. In particular, local client may not have access to all the domains, and their domains may only overlap partially; in the extreme case, local data included in one client may naturally form a unique domain. Therefore, whether centralized generalization ability is achievable when local client has heterogeneous domains remains unknown. Further, if some distributed algorithms can achieve the centralized generalization ability as if the server has information of all the domains, the communication cost remains unclear. In addition, the impact on the generalization ability of the total clients number remains unknown.

**Adapting Centralized DG Methods to FL Setting.** To adapt centralized methods, we simply run the centralized DG method at each client locally with their own local dataset (see subsection 3.1 for how the local datasets are created), and then compute an average of model parameters at each communication round (see next paragraph). This approach is straightforward for the homogeneous ($\lambda = 1$) and heterogeneous ($\lambda = 0.1$) settings where each client has data from all domains— albeit quite imbalanced for $\lambda = 0.1$. This can be seen as biased updates at each client based on biased local data. Similarly, this approach can be implemented in the domain separation case *if* at least one client holds *multiple* non-overlapping domains (i.e., $\exists k, |P_k| \geq 2$, which would happen if $K < M$). However, if all clients only have one primary domain, i.e., $\forall k, |P_k| = 1$, which will happen if $K \geq M$, this simple approach cannot be extended to the domain separation setting ($\lambda = 0$) because centralized DG methods require data from at least two domains. In fact, these centralized DG methods degenerate to ERM if there is only one domain per client. Extending these methods to the case where all clients only have one domain is an interesting direction for future work.

**Synchronization Schedule and Batch Creation** For all algorithms, we run $E$ epochs locally on each client and then the server computes a weighted average of the resulting models, i.e., $\theta^{t+1} = \sum_{k=1}^K \frac{n_k}{n} \theta_k^{t+1}$. Because each epoch runs through the whole dataset, the $k$-th client runs $\sum_m n_{m,k}/B$ batches, where $B$ denotes the batch size. For FedAvg (ERM) and FedGMA, we uniform randomly sample a batch from the local dataset without considering the domain labels. For the FL adaptations of centralized algorithms, we use the sampling method from the WILDS benchmark Koh et al. (2021), namely each client uniform randomly samples two domains from its local dataset and then uniform randomly samples $B/2$ examples from each domain without replacement. Beyond simple model averaging, FedGMA Tenison et al. (2022) additionally adopts a global masking operation at the server for update changes. We refer the readers to Appendix B.2 for the detailed hyperparameters including learning rate, batch size, and model selection.

## 4 MAIN RESULTS

We adapt six representative centralized DG methods into FL context, include IRM Arjovsky et al. (2019), Fish Shi et al. (2021), MMD Li et al. (2018), Coral Sun & Saenko (2016), GroupDRO Sagawa et al. (2019), Mixup Zhang et al. (2017), and compare them with FedDG Liu et al. (2021), FedADG Zhang et al. (2021), FedSR Nguyen et al. (2022) and FedGMA Tenison et al. (2022) which are naturally designed for solving domain generalization tasks in federated learning.

### 4.1 BASELINE SETTING (PACS AND FEMNIST-62)

In the baseline setting, we consider three domain heterogeneity regimes on image classification tasks PACS Li et al. (2017) widely used in domain generalization, and on FEMNIST-62 (digits and characters) Cohen et al. (2017), a prototype dataset in the FL context.

**PACS.** We evaluate PACS with $K = 100$ clients, $M = 2$ training domains (cartoon, sketch), 1 validation domain (art painting) and 1 test domain (photo). The maximum communication rounds is set to 50 with 1 local epoch per communication. The domain balance parameter was varied $\lambda \in \{1, 0.1, 0\}$. For the domain separation case $\lambda = 0$, each client locally only has *one* training domain (i.e., $\forall k, |P_k| = 1$); in this case, no centralized methods are suitable (see subsection 3.2) so we only compare the four federated domain generalization methods to ERM.

Figure 2 plots the held-out test accuracy versus communication rounds on PACS with increasing domain heterogeneity. As seen from Figure 2, **1)** most algorithms perform reasonably well in the homogeneous case ($\lambda = 1$) except FedADG and FedSR. In fact those two fail in all three cases. They are sensitive to the client number and work favorably when $K$ is small, e.g. $K = 2$, see subsection 4.2 for discussion on the effect of number of clients. **2)** As domain heterogeneity increases, i.e., $\lambda$ from 1 to 0.1, the algorithms consistently converge slower and have worse test accuracy, which demonstrates that domain heterogeneity among clients is a unique extra challenge introduced by FL. In particular, the centralized DG methods FISH, CORAL, IRM, and MMD extended to the FL setting have poor performance compared with that of ERM in the heterogeneous case, while Group-DRO outperforms ERM both in homogeneous and heterogeneous case. **3)** In the domain separation case, because $K > M$ for PACS, each client locally only holds one training domain, and thus no centralized methods are suitable to use. FedDG requires sharing the amplitude spectrum of images among local clients, which causes privacy concerns. Therefore, even for this dataset containing only four domains in total, prior works struggle to compete with ERM.

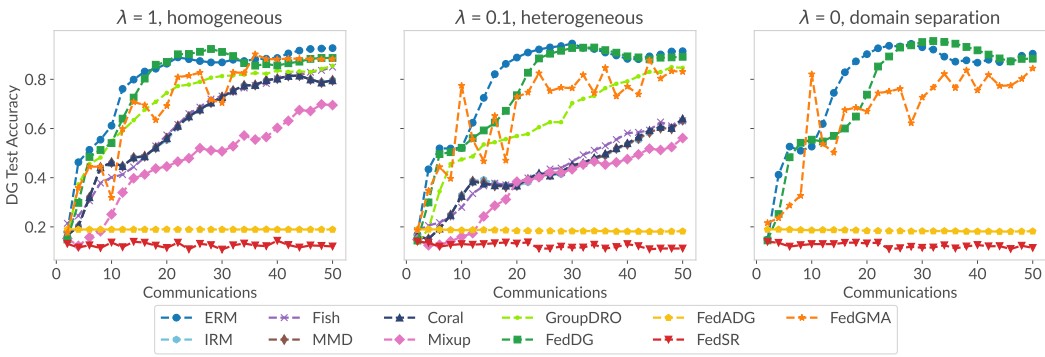

Figure 2: Accuracy versus communication rounds for PACS; total clients and training domains $(K, M) = (100, 2)$; increasing domain heterogeneity from left to right: $\lambda = (1, 0.1, 0)$.

We also report the final test accuracy using held-out-domain validation in Table 2, similar results with in-domain validation is deferred to Table. 13 in Sec.Appendix C. It shows as the domain heterogeneity increases ($\lambda$ from 1 to 0), the test accuracy of current centralized methods (except GroupDRO) degrades dramatically using either validation principle. For the current four natural FL algorithms, only FedDG works comparably to ERM, FedGMA performs worse than the above two on PACS, and because FedADG and FedSR are sensitive to the number of clients, they both fail to converge in this case as $K = 100$.

Table 2: Test accuracy on PACS dataset with held-out-domain validation; total client number $K = 100$; total training domain number $M = 100$. N.A. refers to not applicable.

| | Centralized | $\lambda = 1$ | $\lambda = 0.1$ | $\lambda = 0$ |
|---|---|---|---|---|
| ERM | 0.8389 | 0.8766 | 0.9144 | 0.9377 |
| IRM | 0.9180 | 0.8156 | 0.5449 | N.A. |
| Fish | 0.9030 | 0.8497 | 0.6311 | N.A. |
| Mixup | 0.8635 | 0.7653 | 0.5551 | N.A. |
| MMD | 0.9186 | 0.8150 | 0.6341 | N.A. |
| Coral | **0.9216** | 0.8150 | 0.5515 | N.A. |
| GroupDRO | 0.9060 | **0.9395** | **0.9437** | N.A. |
| FedDG | 0.8922 | 0.9234 | 0.9275 | **0.9521** |
| FedADG | 0.8922 | 0.1892 | 0.0592 | 0.0598 |
| FedSR | 0.8754 | 0.1246 | 0.1263 | 0.1257 |
| FedGMA | N.A. | 0.882 | 0.8467 | 0.8446 |

**FEMNIST-62 (digits and characters).** We consider the FL prototyping dataset FEMNIST-62 with $K = 100$ clients and treat handwritten digits and characters each users created as a natural domains. We split into $M = 2586$ training domains, 320 validation domains, and 331 test domains. The maximum communication rounds is set to 50 with 1 local epoch per communication. The domain balance parameter is varied $\lambda \in \{1, 0.1, 0\}$. For the homogeneous and heterogeneous cases, each client locally holds $M = 2586$ training domains. The data are divided according to Equation 1.

For the domain separation case, given that $K < M$, we use Algorithm 1 to split the domains to each client. Therefore, each client holds multiple non-overlapping domains. All of the methods are applicable on this dataset. We observe that as the domain balance parameter $\lambda$ decreases from 1 to 0, FedGMA, FedSR are consistently comparable to ERM while the others fail. The in-domain and held-out domain accuracy are reported in Table 7, Sec.Appendix C.

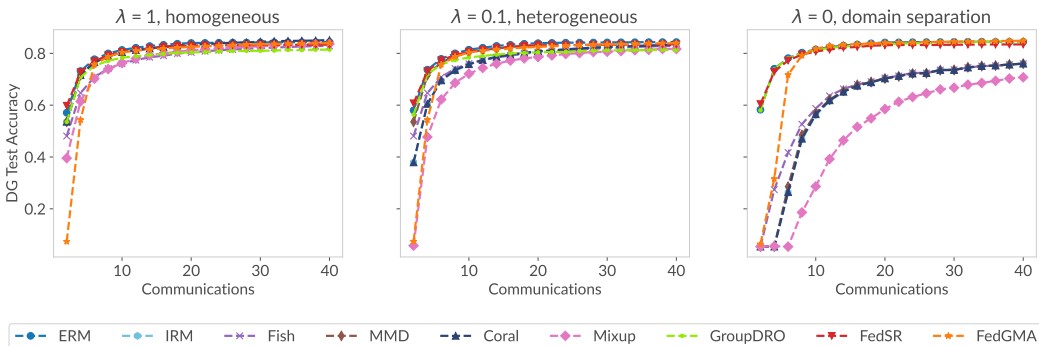

Figure 3: Accuracy versus communication rounds for FEMNIST; total clients and training domains $(K, M) = (100, 2586)$; increasing domain heterogeneity from left to right panel: $\lambda = (1, 0.1, 0)$.

## 4.2 More Realistic Datasets (Py150, CivilComments, iWildCam)

In this subsection, we tackle more realistic domain generalization tasks in the FL context. We include three domain heterogeneity regimes on natural language processing dataset Py150 (with $8421$ domains), CivilComments (with $16$ domains), and image dataset iWildCam (with $323$ domains). Note Mixup Zhang et al. (2017), FedDG Liu et al. (2021), and FedADG Zhang et al. (2021) are only suitable for image datasets, thus are excluded in the experiments on Py150 and CivilComments.

In the following experiments on these three datasets, we choose maximum communication round $C = 50$ for IWildCam, and $C = 10$ for Py150 and CivilComments, 1 local epoch per communication.

Table 3: Test accuracy on IWildCam dataset with held-out-domain validation; total client number $K = 243$, training domain $M = 243$.

| | Centralized | $\lambda = 1$ | $\lambda = 0.1$ | $\lambda = 0$ |
|---|---|---|---|---|
| ERM | 0.2727 | **0.1762** | **0.1241** | **0.0707** |
| IRM | 0.2629 | 0.0718 | 0.0903 | N.A. |
| Fish | 0.3103 | 0.0781 | 0.0977 | N.A. |
| Mixup | 0.2923 | 0.0264 | 0.0370 | N.A. |
| MMD | 0.3134 | 0.0718 | 0.0924 | N.A. |
| Coral | **0.3270** | 0.0718 | 0.0926 | N.A. |
| GroupDRO | 0.2531 | 0.0530 | 0.0908 | N.A. |
| FedDG | 0.2771 | 0.1445 | 0.1191 | 0.0503 |
| FedADG | 0.0049 | 0.0049 | 0.0049 | 0.0049 |
| FedSR | 0.0056 | 0.0056 | 0.0056 | 0.0056 |
| FedGMA | N.A. | 0.0106 | 0.0957 | 0.0106 |

communication. For the homogeneous and heterogeneous case, respectively, each client locally holds all the training domains. For the domain separation case, we use Algorithm 1 to split the domains to each client. When $K < M$, client locally holds non-overlapping domains; all of the centralized methods are applicable in this case. When $K \geq M$, each client locally only holds one training domain; thus, no centralized method is applicable, only four natural federated domain generalization methods are applicable, we compare them with ERM.

**Py150:** We evaluate Py150 with 100 clients, 5477 training domains, 261 validation and 2471 test domains. Given that $K < M$, we compare all the methods except Mixup, FedDG and FedADG in all three different domain heterogeneity regimes. **IWildCam:** We evaluate IWildCam with 323 clients, 243 training domains, 32 validation and 48 test domains. We compare all the methods in homogeneous and heterogeneous regimes. Given that $K = M$, we can only compare FedDG, FedADG, FedSR, FedGMA with ERM in the domain separation regime. **CivilComments:** CivilComments is a special kind of DG where the test domain is a subpopulation of the training domain, and our goal is to perform well on the worst-case domain. CivilComments contains 100 clients and 16 domains. We compare all the methods except Mixup, FedDG and FedADG in homogeneous and heterogeneous regimes. Given that $K < M$, we can only compare FedSR, FedGMA with ERM in the domain separation regime.

Table 4: Test accuracy with held-out-domain validation; total client number $K = 100$.



(a) Py150 Dataset with $M = 5477$

| | Centralized | $\lambda = 1$ | $\lambda = 0.1$ | $\lambda = 0$ |
|---|---|---|---|---|
| ERM | **0.6833** | **0.6827** | **0.3874** | 0.3343 |
| IRM | 0.6771 | 0.6722 | 0.374 | **0.3734** |
| Fish | 0.6634 | 0.6731 | 0.3734 | 0.2487 |
| MMD | 0.6558 | 0.6731 | 0.3745 | 0.3039 |
| Coral | 0.6558 | 0.6723 | 0.3755 | 0.3325 |
| GroupDRO | 0.5127 | 0.6824 | 0.3358 | 0.3356 |
| FedSR | | | | |
| FedGMA | N.A. | 0.6628 | 0.3737 | 0.3567 |

(b) CivilComments Dataset with $M = 16$

| | Centralized | $\lambda = 1$ | $\lambda = 0.1$ | $\lambda = 0$ |
|---|---|---|---|---|
| ERM | 0.5413 | 0.3929 | 0.3390 | **0.3331** |
| IRM | 0.6408 | 0.6153 | **0.6093** | N.A. |
| Fish | **0.6713** | **0.6435** | 0.6086 | N.A. |
| MMD | 0.6520 | 0.6121 | **0.6093** | N.A. |
| Coral | 0.5847 | 0.6121 | **0.6093** | N.A. |
| GroupDRO | 0.6383 | 0.6028 | 0.4954 | N.A. |
| FedSR | 0.6442 | 0.3603 | 0.3389 | 0.3188 |
| FedGMA | N.A. | 0.3341 | 0.3157 | 0.3195 |



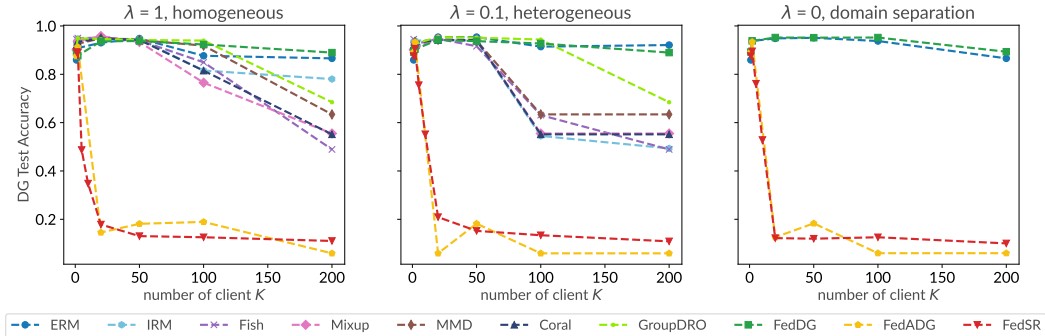

Figure 4: PACS: Held-out DG test accuracy versus number of clients.

We compare the final test accuracy using held-out validation in Table 4a for Py150, Table 4b for CivilComments, and Table 3 for IWildCam. (Using two validation criterion are summarized in Table 8, Table 9 and Table 10 in Appendix C). The results show that the performance of ERM dominates the other algorithms on these three datasets. No algorithm achieves its centralized domain generalization ability, where centralized corresponds to training on the centralized dataset gathered from all the training clients. We also plot accuracy versus communication figures in Figure 7 for Py150, Figure 6 for CivilComments and Figure 8) for IWildCam in Sec. C.

### 4.3 ADDITIONAL FL-SPECIFIC CHALLENGES FOR DOMAIN GENERALIZATION

Besides domain heterogeneity, we also investigate the challenges brought by FL, including massive number of clients number and communication constraints, which are unique to the FL setting.

**i) Massive number of clients:** In this experiment, we explore the performance of different algorithms when the number of clients $K$ increases on PACS. We fix the communication rounds $C = 50$ and the local number of epoch is $1$ (synchronizing the models every epoch). Figure 4 plots the held-out DG test accuracy versus number of clients for different levels of data heterogeneity. The following comments are in order: given communication budget, 1) current domain generalization methods all degrade a lot in particular after $K \geq 50$, while the performance ERM and FedDG maintain relatively unchanged as the clients number increases given communication budget. FedADG and FedSR are are sensitive to the clients number, and they both fail after $K \geq 20$. 2) Even in the simplest homogeneous setting $\lambda = 1$, where each local client has i.i.d training data, current domain generalization methods IRM, FISH, Mixup, MMD, Coral, GroupDRO work poorly in the existence of large clients number, this means new methods are needed for DG in FL context when data are stored among massive number of clients.

**ii) Communication constraint:** To show the effect of communication rounds on convergence, we plot the test accuracy versus communication rounds in Figure 5. We fix the number of clients $K = 100$ on PACS and decreases rounds of communication (together with increasing local epochs), that is, $C = (50, 10, 5)$ (with $E = (1, 5, 10)$). That is, if the regime restricts the communication budget, then we increase its local computation $E$ to have the same the total computations. Therefore, the influence of communication on the performance is fair between algorithms because the total

data pass is fixed. We observe that 1) the curves are relatively "flat" for most of algorithms, this is predictable because we vary the number of pass of the data $E$ per round for a changing $C$, and locally the aggregation rules are the same $\theta^{t+1} = \sum_{k=1}^{K} \frac{n_k}{n} \theta_k^{t+1}$, where $n_k$ is the training sample size at client $k$. 2) In particular, ERM, GroupDRO, and FedDG can achieve comparable good performance when communication budget is low ($C = 10$) comparable to when there communication budget is high ($C = 50$), showing their communication efficiency in the FL context.

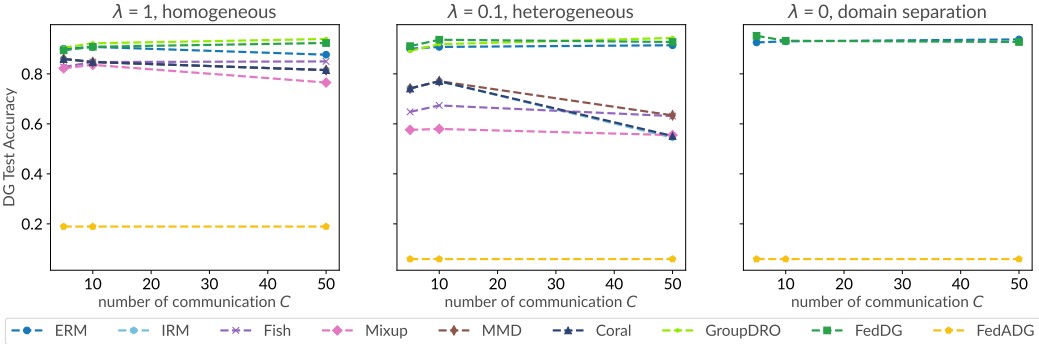

Figure 5: PACS: Held-out DG test accuracy vs. varying communications (resp. varying echoes ).

## 5 CONCLUSION AND DISCUSSION

This work evaluates multiple algorithms to solve domain generalization task in the federated learning context. We evaluate the influence of client domain heterogeneity, the total number clients, and communication rounds on the domain generalization ability. We show that DG in FL context is an *unsolved* problem, and it brings abundant new challenges. Specifically, the following aspect might be future directions of interest: 1) domain heterogeneity across the clients dramatically impacts the generalization ability on image dataset PACS and IWildCAM as well as NLP datasets Py150 and CivilComments; designing new FL algorithm to recover centralized domain generalization ability remains open; 2) previous works (eg, FedSR, FedADG) evaluate the generalization ability built upon small number of clients $K$ is not enough, massive clients setting needs to be take into consideration; 3) more realistic datasets need to be considered in the domain generalization in FL context; 4) For the domain separation case, few prior works are applicable to the case where each client only holds one domain—new DG algorithms for the FL setting are required for this case. We list the gap table below for summarizing the current DG algorithms performance gap w.r.t ERM in the FL context, in particular, positive means it outperforms ERM, negative means it is worse than ERM. It can be seen that in the on the simple dataset, the best DG migrated from centralized setting is better than ERM. In the domain separation case, no centralized DG algorithms can be adapted to it, and FedDG and FedADG performs comparably good in this setting. However, they fail in harder datasets. Federated DG algorithms that outperforming ERM, supporting NLP dataset, and free of data sharing are still in need.

Table 5: Gap Table: Current Progress in solving DG in FL context

| | Comparing to FedAvg-ERM | | | | |
| --- | --- | --- | --- | --- | --- |
| | DG migrated from centralized setting | FedDG | FedADG | FedSR | FedGMA |
| Domain Separation | ✗ | ✓ | ✓ | ✓ | ✓ |
| Baseline Dataset: PACS | +2.93% | +1.44% | −87.99% | −81.20% | −9.32% |
| Baseline Dataset: FEMNIST | +5.14% | | | −0.93% | −0.28% |
| Realistic Dataset: Py150 | +3.91% | ✗ | ✗ | | +2.24% |
| Realistic Dataset: CivilComments | +27.03% | ✗ | ✗ | −1.43% | −1.36% |
| Realistic Dataset: IWildCam | -10.44% | −3.17% | −2.04% | −6.51% | −6.26% |
| Free of Data Sharing | ✓ | ✗ | ✓ | ✓ | ✓ |

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

## A    REPRODUCIBILITY STATEMENT

Code for reproduce the result is available at the anonymous link. We include detailed documentation in using our code to reproduce the results throughout the paper. We also provide documentation in adding new algorithm's DG evaluation in the FL context.

## B    EXPERIMENTS SETTING SUPPLEMENTARY

### B.1    DATASETS AND MODELS

In this section, we introduce the datasets we used in our experiments, and the split method we used to build heterogeneous datasets in the training and testing phase as well as the heterogeneous local training datasets among clients in the FL. **Datasets.** We contain three datasets as the benchmark: PACS Li et al. (2017), IWildCam Koh et al. (2021), and Py150 Koh et al. (2021). These three datasets cover different levels of difficulty as well as different types of tasks. PACS and IWildCam are both image classification datasets and Py150 is a natural language processing (NLP) dataset. PACS is an image dataset for domain generalization. It consists of four domains, namely Photo (1,670 images), Art Painting (2,048 images), Cartoon (2,344 images), and Sketch (3,929 images). This task requires learning the classification task on a set of objects by learning on totally different renditions. The Py150 is a natural language processing dataset containing $150,000$ python source code dataset from $8,421$ repository. The goal is to predict the next token given the context of previous tokens. This is a real-world NLP dataset that contains multiple repositories which naturally form multiple domains. The IWildCam contains wild animals captured by multiple heats or motion-activated static cameras. Due to the variation in camera model, position, color, background, and relative animal frequencies, the samples form multiple domains. It contains $203,029$ images from 323 different camera traps, the images contain 182 different animal species. For Py150 and IWildcam datasets, we follow the same split method as the Wilds Koh et al. (2021). For PACS, we use cartoon and sketch as the training domains, art-painting as the held-out-validation domain, and photo as the test domain, and we use $90\%$ of the data from cartoon and sketch domains to be the training domains, and about $5\%$ to be in-domain validation, and other $5\%$ to be the in-domain test set. During the sampling, we keep the class distribution the same among the training dataset, the in-domain validation dataset, and the in-domain test dataset. **Models.** For image classification datasets PACS and IWildCam, we use ResNet50 model He et al. (2016), and Py150 is a natural language processing (NLP) dataset where we use OpenAI GPT2 Radford et al. (2019) to train.

### B.2    HYPERPARAMETERS AND MODEL SELECTION

**Hyperparameters** To make fair comparisons, we allocate the same budget during training for each algorithm on each dataset. The budget includes the times of allowed hyperparameter search, model architecture, local computation resources, and communication rounds. For each dataset, we fix the model architecture and initialization to be the same. We conduct eight times hyperparameter searching for each algorithm, and choose the set of hyperparameters that achieves the best performance. For all the datasets and algorithms, we set $100\%$ for clients' participation in the training during each communication.

For PACS, we fix the number of clients $K = 100$, 50 communications in total where each communication happens after one epoch of local training. the batch size is 64. We use Adam optimizer and the learning rate is $1 \times 10^{-3}$ except for FedSR algorithm, where we choose SGD with lr= 0.002, momentum= 0.9 and weight decay= $5 \times 10^{-4}$.

For Py150, we set $K = 100$, and the total number of communication is 3 in the centralized case and 10 in the distributed case. The batch size is 96. We use AdamW optimizer, lr = $8 \times 10^{-5}$, and $\epsilon = 1 \times 10^{-8}$.

For IWildcam, the number of clients $K = 243$, 12 communications in the centralized case, and 50 communications in the federated learning case, and the batch size is 16. We use Adam optimizer where the learning rate is $3 \times 10^{-5}$.

For FEMNIST, the number of clients $K = 100$, 20 communications in the centralized case, and 40 communications in the federated learning case, and the batch size is 64. We use Adam optimizer where the learning rate is $1 \times 10^{-3}$.

For CivilComments, the number of clients $K = 100$, 5 communications in the centralized case, and 10 communications in the federated learning case, and the batch size is 16. We use Adam optimizer where the learning rate is $1 \times 10^{-5}$.

For each dataset, we choose the hyperparameters starting from their default value consistent with the choice in previous benchmarks DomainBed Gulrajani & Lopez-Paz (2020) Wilds Koh et al. (2021), or their value proposed in the original paper .

**Model Selection**   In DG, model selection could significantly affect the model performance. During training, we evaluate the aggregated model by using the validation dataset after each communication round. After performing $C$ communication rounds, we select the model that achieves the best performance on the validation dataset. This work adopts two model selection methods for the DG task: in-domain and held-out-domain model selection Gulrajani & Lopez-Paz (2020). In-domain model selection method uses validation dataset which is independent and identically sampled from the training dataset. The held-out-domain model selection method uses validation dataset that only contains examples from a set of domains that do not overlap with the training and testing domains.

## C   OTHER EVALUATION

We put our extra evaluations here for reference. The experiments are summarized in terms of dataset.

### C.1   **PACS and FEMNIST**

Table 6: Test accuracy on PACS with two model selection criterion: in-domain / held-out domain validation; total clients number $K = 2$. First column "centralized" corresponding to the centralized domain generalization accuracy, i.e., client number $K = 1$. Increasing domain heterogeneity from left panel to right panel: $\lambda = (1, 0.1, 0)$. The number in boldface highlights the highest test accuracy in that column. N.A. refers to not applicable.

|  | In-Domain Validation | | | | Held-Out-Domain Validation | | | |
|---|---|---|---|---|---|---|---|---|
|  | Centralized | $\lambda = 1$ | $\lambda = 0.1$ | $\lambda = 0$ | Centralized | $\lambda = 1$ | $\lambda = 0.1$ | $\lambda = 0$ |
| ERM | 0.8389 | 0.8982 | 0.8234 | 0.8958 | 0.8593 | 0.9084 | 0.9144 | 0.9377 |
| IRM | 0.918 | 0.9174 | 0.8814 | N.A. | 0.9347 | 0.9467 | **0.9389** | N.A. |
| Fish | 0.8916 | **0.9359** | 0.9204 | N.A. | **0.9449** | **0.9497** | 0.9251 | N.A. |
| Mixup | 0.9060 | 0.9371 | 0.9072 | N.A. | 0.9012 | 0.9371 | 0.9072 | N.A. |
| MMD | **0.9072** | 0.9299 | 0.8928 | N.A. | 0.9210 | 0.9263 | 0.9335 | N.A. |
| Coral | 0.8886 | 0.9138 | 0.8988 | N.A. | 0.9204 | 0.9341 | 0.9311 | N.A. |
| GroupDRO | 0.8952 | 0.9018 | 0.8964 | N.A. | 0.9150 | 0.9491 | 0.9269 | N.A. |
| FedDG | 0.9024 | 0.8701 | 0.8683 | 0.6856 | 0.8922 | 0.8772 | 0.9036 | **0.9383** |
| FedADG | 0.9024 | 0.7958 | **0.9569** | **0.9473** | 0.8922 | 0.9126 | 0.9329 | 0.9341 |
| FedSR | 0.8246 | 0.8725 | 0.9060 | 0.8928 | 0.8617 | 0.8898 | 0.9060 | 0.8952 |
| FedGMA | 0.8485 | 0.8850 | 0.7922 | 0.8114 | 0.8602 | 0.8850 | 0.9425 | 0.9251 |

We report the final test accuracy using two validation criterion in Table 6. It shows that the homogeneity case ($\lambda = 1$ column) may even slightly outperform the its counterpart centralized domain generalization accuracy in the simple case (small client and domain numbers). This could come from the natural regularization brought by the FL. In the domain separation case ($\lambda = 0$ column), although FedADG is not communication efficient as shown in Fig 2, it seems to be more robust across validation strategies whereas FedDG performs poorly using in-domain validation.

Overall, as expected, existing DG algorithms with enough communication rounds are able to perform reasonably well in this simple setting where the dataset is simple and the number of clients and domains are small.

Table 7: Test accuracy on FEMNIST dataset with two model selection criterion: in-domain / held-out-domain validation; total client number $K = 100$; total training domain number $M = 3500$.

|  | In-Domain Validation | | | | Held-Out-Domain Validation | | | |
|---|---|---|---|---|---|---|---|---|
|  | Centralized | $\lambda = 1$ | $\lambda = 0.1$ | $\lambda = 0$ | Centralized | $\lambda = 1$ | $\lambda = 0.1$ | $\lambda = 0$ |
| ERM | 0.8544 | 0.8368 | 0.8398 | 0.8399 | 0.8544 | 0.8368 | 0.8398 | 0.8399 |
| IRM | 0.8442 | 0.8303 | 0.8389 | 0.8328 | 0.8442 | 0.8303 | 0.8389 | 0.8328 |
| Fish | 0.8487 | 0.8331 | 0.8290 | 0.8264 | 0.8487 | 0.8331 | 0.8290 | 0.8264. |
| Mixup | 0.8337 | 0.8275 | 0.8168 | 0.8161 | 0.8337 | 0.8275 | 0.8168 | 0.8161 |
| MMD | 0.8442 | 0.8430 | 0.8323 | 0.8293 | 0.8442 | 0.8430 | 0.8323 | 0.8293 |
| Coral | 0.8457 | 0.8403 | 0.8328 | 0.8513 | 0.8457 | 0.8403 | 0.8328 | **0.8513** |
| GroupDRO | 0.8413 | 0.8447 | 0.8139 | 0.8052 | 0.8413 | 0.8447 | 0.8139 | 0.8052 |
| FedDG |  |  |  |  |  |  |  |  |
| FedADG |  |  |  |  |  |  |  |  |
| FedSR | 0.8470 | 0.8320 | 0.8316 | 0.8306 | 0.8470 | 0.8300 | 0.8316 | 0.8306 |
| FedGMA | N.A. | 0.8486 | 0.8395 | 0.8371 | N.A. | 0.8486 | 0.8395 | 0.8371 |

## C.2 MORE REALISTIC DATASETS (PY150, CIVILCOMMENTS, IWILDCAM)

To observe the convergence of each algorithm, we plot Figure 7 for Py150 (resp. Figure 6 for Civil-Comments and Figure 8 for IWildCam). It shows that with realistic dataset as well as with non-trivial number of clients, all of the algorithms tend to be more sensitive to domain heterogeneity. Even in the heterogeneous case, where each client locally holds all the training domains, their generalization abilities on the unseen domains are much worse than its centralized counterpart; let alone to solve the even harder domain separation case.

We also reported accuracy using the in-domain and held-out domain validation for Py150 in Table 8, CivilCommnets in Table 9, and IWildCAM in Table 10.

Table 8: Test accuracy on Py150 dataset with two model selection criterion: in-domain / held-out-domain validation; total client number $K = 100$.

|  | In-Domain Validation | | | | Held-Out-Domain Validation | | | |
|---|---|---|---|---|---|---|---|---|
|  | Centralized | $\lambda = 1$ | $\lambda = 0.1$ | $\lambda = 0$ | Centralized | $\lambda = 1$ | $\lambda = 0.1$ | $\lambda = 0$ |
| ERM | **0.6743** | **0.6827** | **0.3874** | 0.3343 | 0.6833 | **0.6827** | **0.3874** | 0.3343 |
| IRM | 0.6771 | 0.6722 | 0.374 | 0.3734 | **0.6771** | 0.6722 | 0.374 | 0.3734 |
| Fish | 0.6634 | 0.6726 | 0.3734 | 0.2487 | 0.6634 | 0.6731 | 0.3734 | 0.2487 |
| MMD | 0.6495 | 0.6731 | 0.3745 | 0.3039 | 0.6558 | 0.6731 | 0.3745 | 0.3039 |
| Coral | 0.6495 | 0.6723 | 0.3755 | 0.3103 | 0.6558 | 0.6723 | 0.3755 | 0.3325 |
| GroupDRO | 0.5127 | 0.6824 | 0.3358 | 0.3356 | 0.5127 | 0.6824 | 0.3358 | 0.3356 |
| FedSR |  |  |  |  |  |  |  |  |
| FedGMA | N.A. | 0.6624 | 0.3737 | 0.3567 | N.A. | 0.6628 | 0.3737 | 0.3567 |

Table 9: Test accuracy on CivilComments dataset with two model selection criterion: in-domain / held-out-domain validation; total client number $K = 100$.

|  | In-Domain Validation | | | | Held-Out-Domain Validation | | | |
|---|---|---|---|---|---|---|---|---|
|  | Centralized | $\lambda = 1$ | $\lambda = 0.1$ | $\lambda = 0$ | Centralized | $\lambda = 1$ | $\lambda = 0.1$ | $\lambda = 0$ |
| ERM | 0.5413 | 0.3929 | 0.3390 | **0.3331** | 0.5413 | 0.3929 | 0.3390 | **0.3331** |
| IRM | 0.6213 | 0.6153 | **0.6093** | N.A. | 0.6408 | 0.6153 | **0.6093** | N.A. |
| Fish | 0.6038 | **0.6435** | 0.6086 | N.A. | **0.6713** | **0.6435** | 0.6086 | N.A. |
| MMD | 0.5981 | 0.6121 | **0.6093** | N.A. | 0.6520 | 0.6121 | **0.6093** | N.A. |
| Coral | 0.5827 | 0.6121 | **0.6093** | N.A. | 0.5847 | 0.6121 | **0.6093** | N.A. |
| GroupDRO | 0.6383 | 0.6028 | 0.4804 | N.A. | 0.6383 | 0.6028 | 0.4954 | N.A. |
| FedSR | **0.6442** | 0.3603 | 0.3389 | 0.3188 | 0.6442 | 0.3603 | 0.3389 | 0.3188 |
| FedGMA | N.A. | 0.3341 | 0.3157 | 0.3195 | N.A. | 0.3341 | 0.3157 | 0.3195 |

Table 10: Test accuracy on IWildCam dataset with two model selection criterion: in-domain / held-out-domain validation; total client number $K = 243$.

| | In-Domain Validation | | | | Held-Out-Domain Validation | | | |
|---|---|---|---|---|---|---|---|---|
| | Centralized | $\lambda = 1$ | $\lambda = 0.1$ | $\lambda = 0$ | Centralized | $\lambda = 1$ | $\lambda = 0.1$ | $\lambda = 0$ |
| ERM | 0.2727 | **0.1762** | **0.1241** | **0.0707** | 0.2727 | **0.1762** | **0.1241** | **0.0707** |
| IRM | 0.2845 | 0.0710 | 0.0903 | N.A. | 0.2629 | 0.0718 | 0.0903 | N.A. |
| Fish | 0.3103 | 0.0781 | 0.0977 | N.A. | 0.3103 | 0.0781 | 0.0977 | N.A. |
| Mixup | 0.2620 | 0.0264 | 0.0371 | N.A. | 0.2923 | 0.0264 | 0.0370 | N.A. |
| MMD | **0.3225** | 0.0711 | 0.0924 | N.A. | 0.3134 | 0.0718 | 0.0924 | N.A. |
| Coral | 0.2857 | 0.0718 | 0.0926 | N.A. | **0.3270** | 0.0718 | 0.0926 | N.A. |
| GroupDRO | 0.2531 | 0.0524 | 0.0908 | N.A. | 0.2531 | 0.0530 | 0.0908 | N.A. |
| FedDG | 0.2866 | 0.1445 | 0.1023 | 0.0503 | 0.2771 | 0.1445 | 0.1191 | 0.0503 |
| FedADG | 0.0049 | 0.0049 | 0.0049 | 0.0049 | 0.0049 | 0.0049 | 0.0049 | 0.0049 |
| FedSR | 0.0056 | 0.0056 | 0.0056 | 0.0056 | 0.0056 | 0.0056 | 0.0056 | 0.0056 |
| FedGMA | N.A. | | | 0.0081 | N.A. | | | 0.0081 |

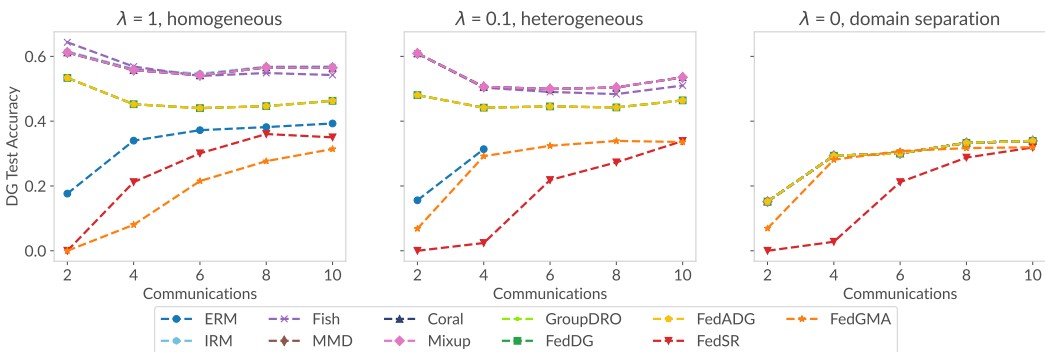

Figure 6: Accuracy versus communication rounds for CivilComments; total clients and training domains $K = 100$ increasing domain heterogeneity from left to right: $\lambda = (1, 0.1, 0)$.

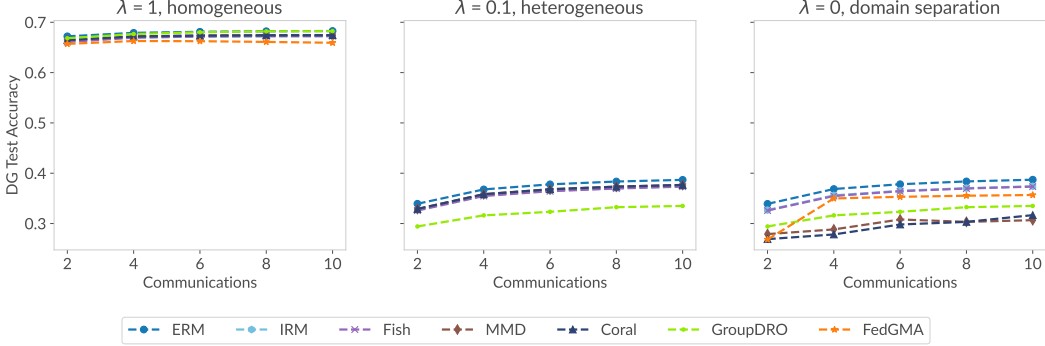

Figure 7: Accuracy versus communication rounds for Py150 ; Total clients number $K = 100$; increasing domain heterogeneity from left to right panel: $\lambda = (1, 0.1, 0)$.

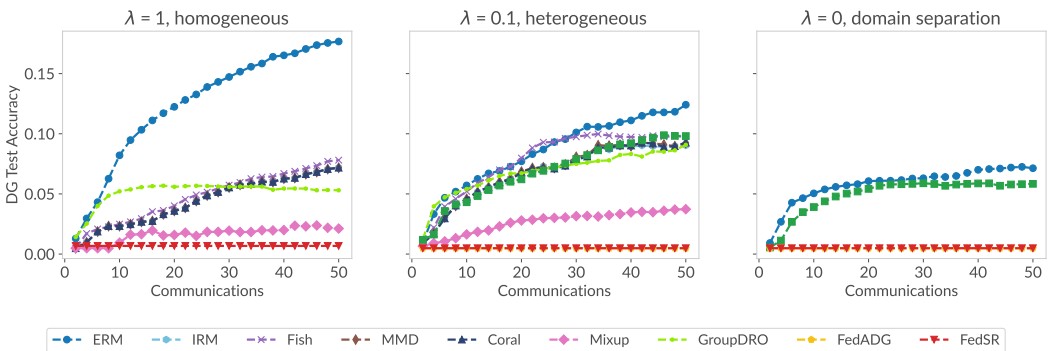

Figure 8: Accuracy versus communication rounds for IWildCam; Total clients number $K = 243$; increasing heterogeneity from left to right panel: $\lambda = (1, 0.1, 0)$.

## C.3 Additional FL-specific challenges for domain generalization

In this subsection, we reported the additional results related to Sec 4.3. Including Table 11, Table 12, Table 13, Table 14 for test accuracy using the held-out and in-domain validation for PACS with $K = 20, 50, 100, 200$, respectively.

Table 11: Test accuracy on PACS dataset with two model selection criterion: in-domain / held-out-domain validation; total client number $K = 20$; total training domain number $M = 2$.

| | In-Domain Validation | | | | Held-Out-Domain Validation | | | |
|---|---|---|---|---|---|---|---|---|
| | Centralized | $\lambda = 1$ | $\lambda = 0.1$ | $\lambda = 0$ | Centralized | $\lambda = 1$ | $\lambda = 0.1$ | $\lambda = 0$ |
| ERM | 0.8389 | 0.9293 | 0.9455 | **0.9569** | 0.8389 | 0.9305 | **0.9539** | 0.9485 |
| IRM | 0.9180 | 0.9461 | 0.9275 | N.A. | 0.9180 | 0.9479 | 0.9431 | N.A. |
| Fish | 0.9030 | 0.9449 | 0.9443 | N.A. | 0.9030 | 0.9515 | 0.9467 | N.A. |
| Mixup | 0.8635 | **0.9581** | 0.9431 | N.A. | 0.8635 | **0.9581** | 0.9461 | N.A. |
| MMD | 0.9186 | 0.9461 | 0.9389 | N.A. | 0.9186 | 0.9491 | 0.9431 | N.A. |
| Coral | **0.9216** | 0.9449 | 0.9461 | N.A. | **0.9216** | 0.9491 | 0.9455 | N.A. |
| GroupDRO | 0.9060 | 0.9401 | **0.9521** | N.A. | 0.9060 | 0.9461 | **0.9539** | N.A. |
| FedDG | 0.9024 | 0.9275 | 0.9275 | 0.8952 | 0.8922 | 0.9365 | 0.9425 | **0.9521** |
| FedADG | 0.9024 | 0.1347 | 0.1269 | 0.1257 | 0.8922 | 0.1455 | 0.1269 | 0.1257 |
| FedSR | 0.8470 | 0.1850 | 0.1886 | 0.1497 | 0.8470 | 0.1784 | 0.2090 | 0.1222 |
| FedGMA | | | | | | | | |

Table 12: Test accuracy on PACS dataset with two model selection criterion: in-domain / held-out-domain validation; total client number $K = 50$; total training domain number $M = 2$.

| | In-Domain Validation | | | | Held-Out-Domain Validation | | | |
|---|---|---|---|---|---|---|---|---|
| | Centralized | $\lambda = 1$ | $\lambda = 0.1$ | $\lambda = 0$ | Centralized | $\lambda = 1$ | $\lambda = 0.1$ | $\lambda = 0$ |
| ERM | 0.8389 | 0.9371 | 0.9437 | **0.9533** | 0.8389 | **0.9479** | **0.9539** | 0.9509 |
| IRM | 0.9180 | 0.9371 | 0.9401 | N.A. | 0.9180 | 0.9455 | 0.9389 | N.A. |
| Fish | 0.9030 | 0.9407 | 0.9030 | N.A. | 0.9030 | 0.9401 | 0.9162 | N.A. |
| Mixup | 0.8635 | **0.9503** | 0.9407 | N.A. | 0.8635 | 0.9335 | 0.9419 | N.A. |
| MMD | 0.9186 | 0.9389 | 0.9329 | N.A. | 0.9186 | 0.9413 | 0.9389 | N.A. |
| Coral | **0.9216** | 0.9389 | 0.9228 | N.A. | **0.9216** | 0.9419 | 0.9413 | N.A. |
| GroupDRO | 0.9060 | 0.9443 | **0.9515** | N.A. | 0.9060 | 0.9419 | 0.9515 | N.A. |
| FedDG | 0.9024 | 0.9329 | 0.9287 | 0.9515 | 0.8922 | 0.9377 | 0.9371 | **0.9515** |
| FedADG | 0.9024 | 0.1862 | 0.1862 | 0.1862 | 0.8922 | 0.1814 | 0.1820 | 0.1832 |
| FedSR | 0.8246 | 0.1269 | 0.1401 | 0.1509 | 0.8754 | 0.1305 | 0.1521 | 0.1275 |
| FedGMA | | | | | | | | |

Table 13: Test accuracy on PACS dataset with two model selection criterion: in-domain / held-out-domain validation; total client number $K = 100$; total training domain number $M = 2$.

|  | In-Domain Validation | | | | Held-Out-Domain Validation | | | |
|---|---|---|---|---|---|---|---|---|
|  | Centralized | $\lambda = 1$ | $\lambda = 0.1$ | $\lambda = 0$ | Centralized | $\lambda = 1$ | $\lambda = 0.1$ | $\lambda = 0$ |
| ERM | 0.8389 | 0.8766 | 0.9144 | **0.8958** | 0.8389 | 0.8766 | 0.9144 | 0.9377 |
| IRM | 0.9180 | 0.8156 | 0.5449 | N.A. | 0.9180 | 0.8156 | 0.5449 | N.A. |
| Fish | 0.9030 | 0.8497 | 0.6311 | N.A. | 0.9030 | 0.8497 | 0.6311 | N.A. |
| Mixup | 0.8635 | 0.7653 | 0.5551 | N.A. | 0.8635 | 0.7653 | 0.5551 | N.A. |
| MMD | 0.9186 | 0.8150 | 0.6341 | N.A. | 0.9186 | 0.8150 | 0.6341 | N.A. |
| Coral | **0.9216** | 0.8150 | 0.5515 | N.A. | **0.9216** | 0.8150 | 0.5515 | N.A. |
| GroupDRO | 0.9060 | **0.9395** | **0.9437** | N.A. | 0.9060 | **0.9395** | **0.9437** | N.A. |
| FedDG | 0.9024 | 0.8868 | 0.8880 | 0.8952 | 0.8922 | 0.9234 | 0.9275 | **0.9521** |
| FedADG | 0.9024 | 0.0915 | 0.0732 | 0.0592 | 0.8922 | 0.1892 | 0.0592 | 0.0598 |
| FedSR | 0.8246 | 0.118 | 0.1341 | 0.1138 | 0.8754 | 0.1246 | 0.1263 | 0.1257 |
| FedGMA |  |  |  |  |  |  |  |  |

Table 14: Test accuracy on PACS dataset with two model selection criterion: in-domain / held-out-domain validation; total client number $K = 200$; total training domain number $M = 2$.

|  | In-Domain Validation | | | | Held-Out-Domain Validation | | | |
|---|---|---|---|---|---|---|---|---|
|  | Centralized | $\lambda = 1$ | $\lambda = 0.1$ | $\lambda = 0$ | Centralized | $\lambda = 1$ | $\lambda = 0.1$ | $\lambda = 0$ |
| ERM | 0.8389 | **0.8635** | 0.8287 | 0.8539 | 0.8389 | 0.8287 | **0.9210** | 0.8659 |
| IRM | 0.9180 | 0.6790 | 0.5000 | N.A. | 0.9180 | 0.7802 | 0.4934 | N.A. |
| Fish | 0.9030 | 0.7802 | 0.4850 | N.A. | 0.9030 | 0.7802 | 0.4892 | N.A. |
| Mixup | 0.8635 | 0.6701 | 0.5551 | N.A. | 0.8635 | 0.6671 | 0.5551 | N.A. |
| MMD | 0.9186 | 0.8150 | 0.6341 | N.A. | 0.9186 | 0.8150 | 0.6341 | N.A. |
| Coral | **0.9216** | 0.8150 | 0.5018 | N.A. | **0.9216** | 0.8150 | 0.4916 | N.A. |
| GroupDRO | 0.9060 | 0.8144 | 0.6090 | N.A. | 0.9060 | 0.8162 | 0.6844 | N.A. |
| FedDG | 0.9024 | 0.7916 | **0.8898** | **0.8868** | 0.9204 | **0.9204** | 0.8898 | **0.8934** |
| FedADG | 0.9024 | 0.0915 | 0.0732 | 0.0592 | 0.8922 | 0.1892 | 0.0592 | 0.0598 |
| FedSR |  |  |  |  |  |  |  |  |
| FedGMA |  |  |  |  |  |  |  |  |

Table 15: Test accuracy on PACS dataset with two model selection criterion: in-domain / held-out-domain validation; total client number $K = 200$; total training domain number $M = 2$; Communication Rounds $C = 5$.

|  | In-Domain Validation | | | | hH-Out-Domain Validation | | | |
|---|---|---|---|---|---|---|---|---|
|  | Centralized | $\lambda = 1$ | $\lambda = 0.1$ | $\lambda = 0$ | Centralized | $\lambda = 1$ | $\lambda = 0.1$ | $\lambda = 0$ |
| ERM | 0.8389 | 0.9006 | **0.9132** | **0.9257** | 0.8389 | 0.9006 | **0.9132** | **0.9257** |
| IRM | 0.9180 | 0.8599 | 0.7431 | N.A. | 0.9180 | 0.8599 | 0.7431 | N.A. |
| Fish | 0.9030 | 0.8275 | 0.6479 | N.A. | 0.9030 | 0.8275 | 0.6479 | N.A. |
| Mixup | 0.8635 | 0.8222 | 0.5760 | N.A. | 0.8635 | 0.8222 | 0.5760 | N.A. |
| MMD | 0.9186 | 0.8605 | 0.7413 | N.A. | 0.9186 | 0.8605 | 0.7413 | N.A. |
| Coral | **0.9216** | 0.8593 | 0.7413 | N.A. | **0.9216** | 0.8593 | 0.7413 | N.A. |
| GroupDRO | 0.9060 | **0.9048** | 0.8916 | N.A. | 0.9060 | **0.9048** | 0.8916 | N.A. |
| FedDG | 0.9024 | 0.8940 | 0.9108 | 0.8952 | 0.8922 | 0.8940 | 0.9108 | **0.9521** |
| FedADG | 0.9024 | 0.0915 | 0.0732 | 0.0592 | 0.8922 | 0.1892 | 0.0592 | 0.0598 |
| FedSR |  |  |  |  |  |  |  |  |
| FedGMA |  |  |  |  |  |  |  |  |

Table 16: Test accuracy on PACS dataset with two model selection criterion: in-domain / held-out-domain validation; total client number $K = 200$; total training domain number $M = 2$; Communication Rounds $C = 10$.

| | In-Domain Validation | | | | Held-Out-Domain Validation | | | |
|---|---|---|---|---|---|---|---|---|
| | Centralized | $\lambda = 1$ | $\lambda = 0.1$ | $\lambda = 0$ | Centralized | $\lambda = 1$ | $\lambda = 0.1$ | $\lambda = 0$ |
| ERM | 0.8389 | **0.9186** | 0.8587 | **0.8958** | 0.8389 | 0.9078 | 0.8970 | 0.9377 |
| IRM | 0.9180 | 0.8347 | 0.7707 | N.A. | 0.9180 | 0.8473 | 0.7707 | N.A. |
| Fish | 0.9030 | 0.8473 | 0.6737 | N.A. | 0.9030 | 0.8473 | 0.6737 | N.A. |
| Mixup | 0.8635 | 0.8359 | 0.5796 | N.A. | 0.8635 | 0.8359 | 0.5796 | N.A. |
| MMD | 0.9186 | 0.8341 | 0.7707 | N.A. | 0.9186 | 0.8473 | 0.7707 | N.A. |
| Coral | **0.9216** | 0.8473 | 0.7707 | N.A. | **0.9216** | 0.8473 | 0.7707 | N.A. |
| GroupDRO | 0.9060 | 0.9222 | 0.9186 | N.A. | 0.9060 | **0.9222** | 0.9186 | N.A. |
| FedDG | 0.9024 | 0.9084 | **0.8880** | 0.8952 | 0.8922 | 0.9084 | **0.9275** | **0.9521** |
| FedADG | 0.9024 | 0.0915 | 0.0732 | 0.0592 | 0.8922 | 0.1892 | 0.0592 | 0.0598 |
| FedSR | | | | | | | | |
| FedGMA | | | | | | | | |

## C.4 TRAINING TIME, COMMUNICATION ROUNDS AND LOCAL COMPUTATION

In this section, we provide training time per communication in terms of the wall clock training time. Notice that for a fixed dataset, most of algorithms have similar training time comparing to ERM, where FedDG and FedADG are significantly more expensive.

Table 17: Wall-clock Training time per communication (unit: s).

| | Wall Clock | | | | |
|---|---|---|---|---|---|
| | PACS | FEMNIST | Py150 | CivilComments | IWildCam |
| ERM | 143 | 262 | 6566 | 3958 | 6301 |
| IRM | 147 | 297 | 7089 | 4085 | 6454 |
| Fish | 148 | 324 | 7770 | 5483 | 7072 |
| Mixup | 144 | 264 | N.A. | N.A. | 6294 |
| MMD | 144 | 287 | 7603 | 4024 | 6663 |
| Coral | 144 | 287 | 7212 | 3901 | 6597 |
| GroupDRO | 145 | 307 | 8121 | 4690 | 9311 |
| FedDG | 352 | | N.A. | N.A. | 32172 |
| FedADG | 181 | | N.A. | N.A. | 11094 |
| FedSR | 151 | 280 | | 4403 | 7136 |
| FedGMA | 143 | 261 | 6545 | 4525 | 6795 |

