# OpenReview forum: "Benchmarking Algorithms for Domain Generalization in Federated Learning"
_ICLR.cc/2023/Conference — Submitted to ICLR 2023_

### Official Review · Reviewer_qSGW · 2022-10-23

**Confidence:** 5
**Correctness:** 3
**Technical Novelty And Significance:** 2
**Empirical Novelty And Significance:** 3
**Recommendation:** 6

**Clarity, Quality, Novelty And Reproducibility:**

The experimental settings for statistical heterogeneity challenge are novel, but the clarity and the organization of the presentation needs to be improved. This paper has open source the code, which seems to be reproducible.


**Strength And Weaknesses:**

Strengths:
1. The DG problem under FL is important and lack of discussion, the proposed benchmark can better promote the research of new algorithms.
2. The experimental settings for statistical heterogeneity challenge under FL is novel and practical.
3. Sufficient experimental results and reproducible code.

Weaknesses:
1. The compared domain generalization methods are old and not representative, which leads to an unconvincing conclusion of this paper.
2. There are lots of FL methods which also focus on the statistical heterogeneity problem and need to be included in the benchmark.
3. The description of proposed experimental setting is not intuitive, making it hard to understand and does not give a clear framework for comparison.


**Summary Of The Paper:**

This paper proposes a new benchmark for federated domain generalization (FedDG) which explores more realistic challenges under the federated learning scenario. The benchmark designs a fair and diverse comparison on the data heterogeneity and communication costs. The experimental results show the challenges on the FedDG stay unsolved in the realistic experiment setting.

**Summary Of The Review:**

The advantages of this work are that the overall benchmark design is innovative and much needed by the academic community, but the disadvantage is that it does not use the recent SOTA method as a comparison, which is insufficient to support the seriousness and necessity of the DG problem under the FL scenario. The lack of comparison of methods from non-iid heterogeneous perspective in the FL field makes the conclusions of this paper less convincing. And the design of the experiment needs to be presented in a clearer and more structured way.

---

> ### Author Response · Authors · 2022-11-09
> **Clarifying questions about other centraized DG methods and FL methods**
>
> Thank you for the comments. While we work towards addressing these comments, we would like to ask a few clarifying questions.
>
> > "The compared domain generalization methods are old and not representative"
>
> We chose  IRM, FISH, MMD, Coral, GroupDRO, and Mixup because they approached domain generalization from *diverse* perspectives (and hence we intended them to be representative of different centralized DG approaches), but more importantly, they are comparable or better than  ERM on the WILDS benchmark leaderboards. We did not intend to comprehensively compare to all centralized domain generalization methods. Rather, we aim to show the gaps of solving DG *in the FL context* as there are already benchmarks for centralized DG. However, we may have missed important representative centralized domain generalization methods that would be competitive in the FL setting and would like to include them in the revision if possible. **Could you suggest some representative centralized methods that we can add in our paper (if time permits)?**
>
> > "There are lots of FL methods which also focus on the statistical heterogeneity problem and need to be included in the benchmark."
>
> We focus on the domain generalization task in FL rather than the general data heterogeneity problem.
> To our best knowledge, most prior FL methods for data heterogeneity still assume the training and testing datasets are the same (i.e., they do not tackle the domain generalization task).
> Thus, they do not seem immediately relevant to our benchmark.
> However, if we have misunderstood or missed prior work, **could you provide references for the FL methods you had in mind?**
> If feasible, we would like to include other relevant methods that we may have missed.

---

> ### Author Response · Authors · 2022-11-19
> **Author Response to Reviewer qSGW**
>
> >"The compared domain generalization methods are old and not representative, which leads to an unconvincing conclusion of this paper."
>
> Thank you for the comment.
> We chose IRM, FISH, MMD, Coral, GroupDRO, and Mixup because they approached domain generalization from diverse perspectives (and hence we intended them to be representative of different centralized DG approaches), but more importantly, they are comparable or better than ERM on the WILDS benchmark leaderboards. We did not intend to comprehensively compare to all centralized domain generalization methods. Rather, we aim to show the gaps of solving DG in the FL context as there are already benchmarks for centralized DG.
>
> In the revision, we have added two new methods federated domain generalization methods FedSR [1] (accepted to the upcoming NeurIPS 2022) and FedGMA [2], both of which were released after the original submission.
>
> However, we may have missed important representative centralized domain generalization methods that would be competitive in the FL setting and would like to include them in the final version. Given that we provided an extensible open-source code library for studying domain generalization in the FL context, new methods are easily to be incorporated into the Benchmark (like FedSR and FedGMA).
> }
>
> > "There are lots of FL methods which also focus on the statistical heterogeneity problem and need to be included in the benchmark."
>
> There are at least two types of realistic statistical data heterogeneity in the FL context.  **Client heterogeneity** is the data heterogeneity between clients involved in training.
> **Train-test heterogeneity** is the data heterogeneity between the training and testing data---e.g., the performance on a new client that was not involved in training or a natural shift in real-world test data due to changes over time, location, or context. In this paper,
> we focus on the domain generalization (i.e., train-test heterogeneity) task in FL rather than the client heterogeneity problem. To our best knowledge, most prior FL methods for data heterogeneity still assume the training and testing datasets are the same (i.e., they do not tackle the domain generalization task). Thus, they do not seem immediately relevant to our benchmark.
>
> > "The description of proposed experimental setting is not intuitive, making it hard to understand and does not give a clear framework for comparison."
>
> Thank you for this constructive comment.
>
> According to your comment, we added a **new figure (Figure 1)** and a **new algorithm (Algorithm 1)** to promote clarity and intuition (see Section 3) on the domain split procedure. We also add **Synchronization Schedule and Batch Creation** in Sec 3.2 for experiments setup clarification.  We also refer the Reviewer to  Appendix B.2 for the detailed hyperparameters including learning rate, batch size, and model selection.
>
> Besides, we have revised  Section  4 to clarify important details.
> To summarize, 1) we regroup together different domain balance parameter $\lambda\in\{1,0.1,0\}$ for each dataset to compare the generalization accuracy on PACS and FEMNIST-62 datasets (see Section 4.1), and on more realistic datasets Py150, CivilComments, iWildCam (see Section 4.2).
> 2) We put the additional FL-specific challenges for domain generalization in Sec 4.3, including massive number of clients and communication constraint.
>
> References
>
> [1] A Tuan Nguyen, Ser-Nam Lim, and Philip HS Torr. Fedsr: Simple and effective domain general- ization method for federated learning.
>
> [2] Tian Li, Anit Kumar Sahu, Manzil Zaheer, Maziar Sanjabi, Ameet Talwalkar, and Virginia Smith. Federated optimization in heterogeneous networks. Proceedings of Machine Learning and Sys- tems, 2:429–450, 2020.

---

> ### Author Response · Authors · 2022-11-30
> **Author Response to Reviewer qSGW | Summary**
>
> We thank the valuable feedback and suggestion from reviewer qSGW. The major concern from reviewer qSGW is that the clarity of the paper, thus we significantly rewrote the paper especially Section 3 and Section 4. We kindly encourage reviewer qSGW to reassess our paper and reach out with any other concerns and questions. We are happy to continue the discussion.

---

### Official Review · Reviewer_oUmB · 2022-10-24

**Confidence:** 3
**Correctness:** 3
**Technical Novelty And Significance:** 2
**Empirical Novelty And Significance:** 3
**Recommendation:** 5

**Clarity, Quality, Novelty And Reproducibility:**

The paper presents the first systematic comparison of the DG-FL methods and as such its analysis is important and it brings new interesting insights. However, the clarity and quality of the paper needs to be significantly improved.

**Strength And Weaknesses:**

Strengths:
-  Domain generalization (DG) in federated learning (FL) is an important problem and has not been a systematic comparison of the methods. This paper presents the first such study.
- The paper presents major challenges that need to be overcome to address DG-FL problems and shows the inability of the existing methods to do so. This is important work needed to advance the field.
- Using homogeneous and heterogeneous splits (as measured by lambda) presents a systematic way to measure how well the algorithm can adapt to heterogeneous  environments.

Weaknesses:
- For benchmark papers, it is extremely important that the paper is well written, well structured and reads well. However, this is not currently the case. There is a lot of information, but the paper does not organize information in a succinct and clear way making it hard to follow. For example, problem setup is never fully defined; part of experiments with varying lambda are in one paragraph, the other in another; it is not clear how centralized DG are extended to FL setting; many tables are not referred to in the paper; it is not explained why centralized DG methods can not be extended to be applicable to lambda=0 setting and so on. These are just a few examples, but the overall impression of the paper is that much more effort is needed to have a well organized and structured paper.
- The methods are tested only on three benchmark dataset. More analysis is needed to make the comparison truly comprehensive
- Similarly, more recent FL algorithms should be included for the comprehensive analysis.

Minor:
- In the paragraph Challenges from more realistic datasets, it should be Fig. 2 (c),
(d) not (b), (c)
- In paragraph 4.1 the authors write that they analyze two values of the lambda parameters, but the figure shows three.




**Summary Of The Paper:**

This paper is focused on benchmarking algorithms for domain generalization (DG) in the federated learning (FL). They evaluate 9 algorithms on three datasets (PACS, iWildCam and Py150). In particular, authors compare centralized DG algorithms and two FL algorithms for DG (FedADG and FedDG). The paper studies the ability of these methods to handle different challenges of DG-FL such as statistical heterogeneity among clients, number of clients, number of communication rounds. The results show more work is needed to design algorithms in the large client setting even for simple datasets as well as to handle local data statistical heterogeneity.

**Summary Of The Review:**

Overall, the analysis presented in the paper is of high importance to the community and has the potential to bring new advances in the DG-FL by outlying unsolved challenges of the current methods. However, more work and effort is needed to make this paper strong and comprehensive which is necessary for these types of benchmarking papers.

---

> ### Author Response · Authors · 2022-11-09
> **Clarifying questions about FL methods and datasets**
>
> We thank the reviewer for the constructive comments on paper organization and will revise the paper per this comment soon, but we would first like to clarify your other comments on datasets and FL algorithms.
>
> > "more recent FL algorithms should be included"
>
> While we do not know of other FL domain generalization works, after we submitted this paper, we came across the concurrent work FedSR [1] accepted to Neurips 2022 that tackles domain generalization in the FL context.
> This very recent concurrent work only compares against one DG+FL method FedADG, which is already in our benchmark.
> We will include this concurrent algorithm in our benchmark in the revised version soon.
> In FL, there are many methods such as FedProx [2] SCAFFOLD [3], and FedSN [4]. However, they are not suitable for solving the domain generalization task because they assume the test dataset and training dataset are from the same domain.
> If we have missed prior work or misunderstood your suggestion, **could you provide references for the "more recent FL algorithms" that you mention?**
>
>
> > "only on three benchmark dataset[sic]...More analysis is needed to make the comparison truly comprehensive"
>
> We plan to include another text dataset CivilComments and a dataset popular in the FL context FEMNIST suggested by another reviewer.
> **Would these datasets alleviate your concern? If not, can you suggest specific datasets that would?**
>
> References
>
> [1] A Tuan Nguyen, Ser-Nam Lim, and Philip HS Torr. Fedsr: Simple and effective domain general-
> ization method for federated learning.
>
> [2] Tian Li, Anit Kumar Sahu, Manzil Zaheer, Maziar Sanjabi, Ameet Talwalkar, and Virginia Smith.
> Federated optimization in heterogeneous networks. Proceedings of Machine Learning and Sys-
> tems, 2:429–450, 2020.
>
> [3] Sai Praneeth Karimireddy, Satyen Kale, Mehryar Mohri, Sashank Reddi, Sebastian Stich, and
> Ananda Theertha Suresh. Scaffold: Stochastic controlled averaging for federated learning. In
> International Conference on Machine Learning, pages 5132–5143. PMLR, 2020.
>
> [4] Brian Bullins, Kshitij Patel, Ohad Shamir, Nathan Srebro, and Blake E Woodworth. A stochastic
> newton algorithm for distributed convex optimization. Advances in Neural Information Process-
> ing Systems, 34:26818–26830, 2021.

---

> ### Author Response · Authors · 2022-11-19
> **Author Response to Reviewer oUmB | Part 1 of 2**
>
> > "For benchmark papers ... needed to have a well organized and structured paper."
>
> We thank the Reviewer for these constructive comments on organizing the paper. We have revised the paper per comment and have done a lot of reorganization in the revised version. Specifically, we kindly ask the Reviewer to look at Section 3 and 4, where in Section 3, we completely rewrite the problem setup and include in detail about **(i)** how to implement domain heterogeneity in the FL context ; and **(ii)** how to adapt centralized methods into the FL context.  In Section 4, as the Reviewer suggested, we have done a significant rewrite for clarity with the new datasets
> and methods to expand upon our original submission. Specifically, In Section 4.1 we put the experiment and analysis for PACS and FEMNIST, given that they are relatively simple tasks. In Section 4.2, we discuss the results for Py150, IWildCAM and CivilComments as they are more realistic datasets. We put the extra realistic challenges brought by FL in Section 4.3 which includes the FL challenges brought by massive clients and communication constraints. Next we address the the Reviewer's concerns one by one.
>
> > "For example, problem setup is never fully defined"
>
> Thank you for this constructive comment.  In Section 3 in the revised version: we first introduce  **(i) how to implement domain heterogeneity in the FL context;** then we introduce the problem of interest **(ii) what is the domain generalization in the federated context;** and **(iii) how to adapt centralized methods into the FL context**.
>
>
>
>
> > "it is not clear how centralized DG are extended to FL setting;"
>
> We thank the Reviewer for pointing out this point, which offers us a chance to clarify the domain separation case. We add a paragraph in the revised version to explain how to deal with the domain separation case. The Reviewer may want to look at Sec. 3.2 under **Adapting Centralized DG Methods to FL Setting** for a detailed discussion.
> We comment it here for the Reviewer's convenience.
> To adapt centralized methods, we simply run the centralized DG method at each client locally with their own local dataset and then compute an average of model parameters at each communication round.
> This approach is straightforward for the homogeneous ($\lambda=1$) and heterogeneous ($\lambda=0.1$) settings where each client has data from all domains---albeit quite imbalanced for $\lambda=0.1$.
> This can be seen as biased updates at each client based on biased local data.
> Similarly, this approach can be implemented in the domain separation case if at least one client holds multiple non-overlapping domains.
>
> However, if all clients only have one  domain, which will happen if $K \geq M$, this simple approach cannot be extended to the domain separation setting ($\lambda = 0$) because centralized DG methods require data from at least two domains.
> In fact, these centralized DG methods degenerate to ERM if there is only one domain per client.
> Extending these methods to the case where all clients only have one domain is an interesting direction for future work.
>
>
>
> > "it is not explained why centralized DG methods can not be extended to be applicable to lambda=0 setting and so on"
>
> Thanks. Please check the previous reply.  We add a paragraph in the revised version Section 3.2 to explain how to deal with the domain separation case  $\lambda=0$.  For the domain separation case, if $K\geq M,$ as mentioned above, the centralized DG methods cannot be extended to the FL context. However, if at least one client holds multiple non-overlapping domains (as would happen when $K<M$). The centralized methods can still be extended to the FL setting, the Reviewer might want to the newly added results in Section 4.1 for EMNIST, and Section 4.2 for Py150, where in both domain separation ($\lambda=0$) case, centralized DG methods can be extended to the FL context.
>
> > "many tables are not referred to in the paper;"
>
> Thanks, fixed. The tables and figures are now organized according to datasets.
>
> > "In the paragraph Challenges from more realistic datasets, it should be Fig. 2 (c), (d) not (b), (c)}"
>
> Thanks, fixed.

---

> ### Author Response · Authors · 2022-11-19
> **Author Response to Reviewer oUmB | Part 2 of 2**
>
> > "The methods are tested only on three benchmark dataset. More analysis is needed to make the comparison truly comprehensive."
>
> Thank you for the comment. We would like to clarify that our current paper is a first step towards comparing different domain generalization methods in the distributed setting. We mainly focused on the extra challenges in domain generalizations in the FL context, i.e., domains and data are scatteted across the clients and there are privacy constraints, communications constraints and the scalability of the total clients number. The aim is to show the gap between centralized DG and DG in the FL context, we do not mean to be exhaustive in the datasets. Indeed,  informed by our findings, this problem remains unsolved even in the baseline setting (please check the results in Section 4.1 for simple PACS dataset with only $4$ domains).
> Therefore, we call for the community suggestions for future works of domain generalization in federated learning.
>
> However, we agree with the Reviewer that it would be good to contain more datasets to show the gap. Besides the original dataset PACS, Py150, iWildcams, we also include another real-world NLP dataset CivilComments [1] and a prototype dataset in the FL context FEMNIST [2] in the revised version.
> We expand our results on these two datasets using the existed methods in the paper as well as two new SOTA FL DG methods, FedSR [3] and FedGMA [4] in the revised version.
>
> > "Similarly, more recent FL algorithms should be included for the comprehensive analysis."
>
> We focus on the domain generalization task in FL rather than the general data heterogeneity problem.
> To our best knowledge, most prior FL methods for data heterogeneity still assume the training and testing dataset are the same (i.e., they do not tackle the domain generalization task).  Thus, they do not seem immediately relevant to our benchmark.
>
> As we mentioned in the above, in the revised version, we add   FedSR and  FedGMA in the revised version.
>
> > "In paragraph 4.1 the authors write that they analyze two values of the lambda parameters, but the figure shows three."
>
> Thank you for the suggestion. We first would like to clarify that in the original version, we separate the three $\lambda$ cases into baseline setting $\lambda=(1,0.1)$ in Section 4.1 and domain separation case $\lambda=0$ in Section 4.2. But we grouped the figures for one dataset (PACS in this case) together. This is not a mistake as the reviewer thought but rather a different style of paper organization. In fact, we are consistent when we comment the results in Section 4.1 and Section 4.2.
>
> However, we agree with the Reviewer, grouping the results for one dataset in one section is easier to follow. Thus, per Reviewer's suggestion, we revised the Section 4 significantly for clarity with the new datasets
> and methods to expand upon our original submission. One of the major changes is regarding this suggestion;  we delete the original domain separation subsection 4.2 and put all three different $\lambda$ cases for PACS together in Sec 4.1; Results of other datasets are also grouped together in order to study the impact of domain balance parameter $\lambda$ on the generalization ability.
>
> > "part of experiments with varying lambda are in one paragraph, the other in another;"
>
> This is a same question as the previous one. Please check the previous reply.
>
>
>
> References
>
> [1] Koh, Pang Wei, et al. "Wilds: A benchmark of in-the-wild distribution shifts." International Conference on Machine Learning. PMLR, 2021.
>
> [2] Caldas, Sebastian, et al. "Leaf: A benchmark for federated settings." arXiv preprint arXiv:1812.01097 (2018).
>
> [3] A Tuan Nguyen, Ser-Nam Lim, and Philip HS Torr. Fedsr: Simple and effective domain general- ization method for federated learning.
>
> [4] Tenison, Irene, et al. "Gradient Masked Averaging for Federated Learning." arXiv preprint arXiv:2201.11986 (2022).

---

> ### Author Response · Authors · 2022-11-30
> **Author Response to Reviewer oUmB | Summary**
>
> We thank the valuable feedback from reviewer oUmB which significantly improves our work. The major concerns from reviewer oUmB are that 1) the paper should be well written; 2) the paper only contains three datasets. Thus, we completely rewrote Section 3 and Section 4 for clarity. We also added a *new figure* and a *new algorithm* to promote clarity (see Section 3). We also added two new datasets and two new algorithms in the revised manuscipt. We encourage reviewer oUmB to reassess our paper in light of our responses and updates. If we adequately address the concern, would it be possible to raise the score?

---

> ### Author Response · Authors · 2022-12-08
> **More Clarifications?**
>
> We thank again for the constructive feedbacks from reviewer oUmB. Here we would like to politely ask reviewer oUmB: Have we addressed your concerns, and if so, would you consider updating the score given the significant changes?
>
> **Summary of Changes**
> 1. Significant revision of Section 3 and Section 4 to promote clarity.
> 2. A new figure (Figure 1) illustrates the data splits method.
> 3. We added two new datasets (FEMNIST and CivilComments) and two new methods (FedSR and FedGMA).
>
> We would like to discuss with you if you need more clarifications.

---

### Official Review · Reviewer_fDMj · 2022-10-24

**Confidence:** 3
**Correctness:** 3
**Technical Novelty And Significance:** 2
**Empirical Novelty And Significance:** 2
**Recommendation:** 6

**Clarity, Quality, Novelty And Reproducibility:**

The clarity and quality is good. This is also the first attempt to benchmark DG in FL. The benchmark should be reproducible as the authors will provide a link to the code and dataset.


**Strength And Weaknesses:**

Strengths

The first paper to benchmark DG in FL settings and considered different setups and difficulties in the benchmark. The list of algorithms included in the benchmark is also comprehensive.
The paper is written clearly and easy to follow.

Weaknesses

Only image datasets were considered in the benchmarks while there are many existing works for DG in the text domain [1,2,3]. Also, only two datasets were considered more realistic which seems to be small as a benchmark.
Some parts of the results are not clear, e.g. why does $\lambda=0.1$ give better performance than $\lambda=1$ for some algorithms in Table 4?

References:

[1] Gururangan, S., Marasović, A., Swayamdipta, S., Lo, K., Beltagy, I., Downey, D., & Smith, N. A. Don’t Stop Pretraining: Adapt Language Models to Domains and Tasks. ACL 2020.

[2] Gururangan, S., Lewis, M., Holtzman, A., Smith, N. A., & Zettlemoyer, L. Demix layers: Disentangling domains for modular language modeling. arXiv 2021.

[3] Chronopoulou, A., Peters, M. E., & Dodge, J.  Efficient hierarchical domain adaptation for pretrained language models. arXiv 2021.


**Summary Of The Paper:**

This paper proposed a benchmark for domain generalization (DG) algorithms in federated learning (FL). The authors considered more realistic settings with larger client numbers and different heterogeneity levels compared to prior works. The benchmarks included two new datasets for DG in FL and multiple existing DG algorithms.


**Summary Of The Review:**

I am leaning towards weak reject as I think the benchmark needs to be more comprehensive as discussed in the weaknesses above.

---

> ### Author Response · Authors · 2022-11-19
> **Author Response to Reviewer fDMj | Part 1 of 2**
>
> > "Only image datasets were considered in the benchmarks''.
>
> We agree with the Reviewer's opinion that text dataset is important to study. However, we kindly disagree with the above assessment.  In fact, we conducted extensively experiments on the next word prediction dataset Py150 [4] and reported the results in Section 4.2 (originally in Section 4.3). The Reviewer might want to look at Figure 7 for the test accuracy versus communication rounds, Table 4(a) / Table 8 for test accuracy on Py150 dataset with held-domain / two model selection criterion and the discussions.
>
> > "there are many existing works for DG in the text domain [1,2,3]"
>
> The Reviewer might want to notice that above three works do not tackle the domain generalization task. In particular,  [1] proposes a second-stage pretraining procedure, and shows that it could improve the model performance for the downstream task. However, this second-stage pretraining needs training samples from the same distribution as the test samples; besides, after this stage, the model still need to be tuned on the training samples which also come from the same distribution as the test samples. Thus, this procedure violates domain generalization (DG) setting where during training, DG could not access data from the test domains.
> [2] proposes a deep model architecture called DEMIX layer. This structure is a collection of feedforward networks, each specialized to a domain, which enable quick domain adaptation. However, the DEMIX layer still need examples from the new domains to fine-tune the model before test, which violates the DG setting.  [3] represents domains as a hierarchical tree where each leaf in the tree represents a specific domain. Similar to DEMIX layer [2], samples from the test domain is still required to form the new leaf in the tree structure.  Though this work is not directly related to the DG. We agree that modeling domain distribution discrepancy is definitely an interesting direction to explore in DG.
>
> > "only two datasets were considered more realistic which seems to be small as a benchmark."
>
> Thank you for the comment. We would like to clarify that our current paper is a first step towards comparing different domain generalization methods in the distributed setting. We mainly focused on the extra challenges in domain generalizations in the FL context, i.e., domains and data are scatters across the clients and there are privacy constraints, communications constraints and the scalability of the total clients number. The aim is to show the gap between centralized DG and DG in the FL context, we do not mean to be exhaustive in the datasets. Indeed,  informed by our findings, this problem remains unsolved even in the baseline setting (please check the results in Section 4.1 for simple PACS dataset with only $4$ domains.)
> Therefore, we call for the community suggestions for future works of domain generalization in federated learning.
>
> However, we agree with the Reviewer that it would be good to contain more datasets to show the gap. Besides the original dataset PACS, Py150, iWildcams, we also include another real-world NLP dataset CivilComments [4] and a prototype dataset in the FL context FEMNIST [5] in the revised version.
> We evaluate IRM, FISH, MMD, Coral, Group DRO, FedSR, FedGMA for CivilComments. Note Mixup, FedDG, and FedADG are only suitable for image datasets, thus are excluded in the experiments on CivilComments. The Reviewer may want to look at Table 4(b) for a test accuracy with held-out-domain validation, and Table. 10 in the Appendix for for test accuracy with in-domain validation. To observe the convergence of each algorithm, we also plot  accuracy versus communication rounds in Figure 6, Section C. Currently, the results on FEMNIST are summarized in Figure 3 for accuracy versus communication rounds in the maintext, and in-domain/ held-out domain accuracy are reported in Table 7, Appendix Sec. C.
>
> References:
>
> [1] Gururangan, S., Marasović, A., Swayamdipta, S., Lo, K., Beltagy, I., Downey, D., & Smith, N. A. Don’t Stop Pretraining: Adapt Language Models to Domains and Tasks. ACL 2020.
>
> [2] Gururangan, S., Lewis, M., Holtzman, A., Smith, N. A., & Zettlemoyer, L. Demix layers: Disentangling domains for modular language modeling. arXiv 2021.
>
> [3] Chronopoulou, A., Peters, M. E., & Dodge, J. Efficient hierarchical domain adaptation for pretrained language models. arXiv 2021.
>
> [4] Koh, Pang Wei, et al. "Wilds: A benchmark of in-the-wild distribution shifts." International Conference on Machine Learning. PMLR, 2021.
>
> [5] Caldas, Sebastian, et al. "Leaf: A benchmark for federated settings." arXiv preprint arXiv:1812.01097 (2018).

---

> ### Author Response · Authors · 2022-11-19
> **Author Response to Reviewer fDMj | Part 2 of 2**
>
> > "Some parts of the results are not clear, e.g. why does   $\lambda=0.1$  give better performance than $\lambda=1$ for some algorithms in Table 4?"
>
> Thank you for this comment, which offers us a chance to clarify the relationship between generalization performance and $\lambda.$ Generally speaking, as the domain balance parameter
>  $\lambda$ decreases, the generalization ability will decrease. For IWildCam, given that it is a complicated dataset, the generalization accuracy of all the algorithms are
> pretty low.  We kindly ask the Reviewer to check Figure 8 in Appendix C.3, where there's a clear trend of performance degradation as $\lambda$ decreases, and it also illustrates how slow the convergence speeds for all the algorithms on IWildCam. The accuracy are still increasing, however with rather slow speeds. We only reported the accuracy after 50 communication rounds given all of the algorithms
> can not converge in a reasonable communication round. Thus, it might be the case for some algorithm, $\lambda=0.1$ has better accuracy than $\lambda=1$ at the round $50.$

---

> ### Author Response · Authors · 2022-11-30
> **Author Response to Reviewer fDMj | Summary**
>
> We thank the time and effort from reviewer fDMj. The major concern from reviewer fDMj is that we do not consider text datasets and only two datasets were considered. Indeed, the original manuscript contains three datasets including a text dataset Py150. In the revision, we added two more datasets with one text-based dataset as well as two more methods. We believe that some parts of the paper might not be clear so reviewer fDMj might misunderstand some information. We thus significantly rewrote Section 3 and Section 4 to promote clarity.
>
> We encourage reviewer fDMj to reassess our paper in light of our responses and updates. If we adequately address the concern, would it be possible to raise the score?

---

> ### Author Response · Authors · 2022-12-08
> **More Clarifications?**
>
> We thank again for the constructive feedbacks from reviewer fDMj, and we would like to politely ask the reviewer: Have we addressed your concerns, and if so, would you consider updating your score given the significant changes?
>
> **Summary of Changes**
> 1. We added two new datasets containing a text-based dataset CivilComments.
> 2. We added two new methods.
> 3. Significant revision of Section 3 and Section 4 to promote clarity.
>
> We would like to discuss with you if you need more clarification.

---

### Official Review · Reviewer_UVoT · 2022-10-27

**Confidence:** 4
**Correctness:** 4
**Technical Novelty And Significance:** 2
**Empirical Novelty And Significance:** 3
**Recommendation:** 6

**Clarity, Quality, Novelty And Reproducibility:**

*Clarity/Quality*
The paper is overall reasonably well written. A clarity issue about how the DG methods are ported to FL is noted.
The reviewer also noted that the appendix is referenced

*Reproducibility*
Code is available

*Novelty*
The benchmark is relevant and the evaluation is more extensive than previous work, but previous work in this topic does exist.


**Strength And Weaknesses:**

Strengths
- The benchmark considers a relevant setting
- The evaluations are detailed covering a variety of methods
- The work highlights the difficulties in this setting

*Weakness*
- The intersection of DG and FL has been considered by several authors recently.. The  following related references and methods are not discussed or compared:
   - Tension et al “Gradient Masked Averaging for Federated Learning”
   - Yuan et al “What do we mean by generalization in Federated Learning”
- The wide array of approaches are appreciated but there are limited details on how these are adapted to the federated setting. Some of these algorithms e.g. MMD, FISH,  seem like they would require access to the different domains at each iteration thus it’s not clear to the reviewer how they are implemented here since clients should not share data.
- Although the performance is evaluated it is not discussed if these approaches come with additional communication or training time overhead. FL is often concerned with these constraints. Related to the previous point are there additional information besides the models transmitted across clients in the various methods?
- The datasets are interesting however in addition to these it might be good to include at least one popular dataset from the FL literature such as FEMNIST (which has been studied in OOD settings in other work) or CIFAR


**Summary Of The Paper:**

The paper proposes a benchmark for Federated Learning in the context of domain generalization. The authors thoroughly evaluate a number of popular DG algorithms on this benchmark.

**Summary Of The Review:**

The paper presents a useful benchmark and highlights the challenges of existing DG methods extended to FL in the context of this benchmark. The evaluation of some existing algorithms is of interest.  Some related work are not fully discussed. The reviewer also currently has some concerns about what data information is transmitted across clients and servers in the methods evaluated.

Update: The authors made  substantial changes to the paper text that have greatly clarified the setting under study, thus have largely addressed my concerns. I am thus increasing my score

---

> ### Author Response · Authors · 2022-11-19
> **Author Response to Reviewer UVoT | Part 1 of 4**
>
> > "The intersection of DG and FL has been considered by several authors recently.. The following related references and methods are not discussed or compared:
> Tension et al "Gradient Masked Averaging for Federated Learning"
> Yuan et al "What do we mean by generalization in Federated Learning''
>
> We thank the Reviewer for pointing out to these two papers. Following her/his suggestion we added in the revised manuscript a literature review of these works; the Reviewer wants to check the discussion under the paragraph  ``Federated Domain Generalization'' in Sec. 2. In addition, we include the Federated Gradient Masking Averaging (FedGMA) into our benchmark in the revised manuscript.
> After we submitted this paper, we came across the concurrent work FedSR [1] accepted to Neurips 2022 that tackles domain generalization in the FL context. This very recent concurrent work only compares against one DG+FL method FedADG, which is already in our benchmark. We have included this concurrent algorithm in our benchmark in the revised version as well.
>
> In particular, we add the evaluations of FedGMA and FedSR into the revised version on the following five datasets: PACS, EMNIST, Py150, CivilComments, and IWildCam. The Reviewer may want to look at revised Section 4, in particular, Figure 2 and Table 2 for PACS, Table 7 for EMNIST, Figure 7 and Table 4(a) for Py150, Figure 6, 8, and  Table 4(b), 3 for CivilComments and IWildCam respectively.  Due to time limits, FedSR on Py150, FedDG and FedADG on FEMNIST are still in progress, which we left as blank in the tables.
>
> To summarize, [2] proposes a new aggregation method called Federated Gradient Masking Averaging (FedGMA) with the goal of improving generalization across clients and of the global model. Their gradient masking prioritizes gradient components that are aligned with the overall dominant direction across clients while the inconsistent components of the gradient are given less importance. Empirically, they show that the OOD generalization ability is higher than the FedAVG. [3] introduced the concept participation gap to identify dataset heterogeneity. They train models using a set of participating clients and examine their performance on held-out data from these clients as well as an additional set of non-participating clients. Therefore it is closely related to domain adaptation in FL context, which is different from our scope where we consider domain generalization ability in the FL context, i.e., the training and test domains do not overlap. [1] proposed FedSR where they enable domain generalization while still respecting the decentralized and privacy-preserving natures of FL context by enforcing $\ell_2$-norm regularizer and a conditional mutual information regularizer on the representation.
>
> References
>
> [1] A Tuan Nguyen, Ser-Nam Lim, and Philip HS Torr. Fedsr: Simple and effective domain general- ization method for federated learning.
>
> [2] Tenison, Irene, et al. "Gradient Masked Averaging for Federated Learning." arXiv preprint arXiv:2201.11986 (2022).
>
> [3] Yuan, Honglin, et al. "What Do We Mean by Generalization in Federated Learning?." International Conference on Learning Representations. 2021.

---

> ### Author Response · Authors · 2022-11-19
> **Author Response to Reviewer UVoT | Part 2 of 4**
>
> >``The wide array of approaches are appreciated but there are limited details on how these are adapted to the federated setting. Some of these algorithms e.g. MMD, FISH, seem like they would require access to the different domains at each iteration thus it’s not clear to the reviewer how they are implemented here since clients should not share data."
>
> Thank you for this constructive comment, which offers us the chance to clarify two points: **(i) how to implement domain heterogeneity in the FL context** ; and **(ii) how to adapt centralized methods into the FL context**. Following the Reviewer's suggestion we added in the revised manuscript.
> more details about how to implement the centralized methods in the revised version; the Reviewer might want to check **Section 3.1 Implementation of domain heterogeneity** and **Section 3.2 Adapting centralized DG methods into the FL context** for details.
>
> **Reply to (i)**  We kindly ask the Reviewer to look at the revised version of Section 3.1, where we provide a concrete  procedure for implementing domain heterogeneity for the benchmark. We summarize here for the Reviewer's convenience. We first provide Algorithm $1$ to split the domains to each client, in a way such that no client shares domains with the others but attempts to balance the total number of training samples between clients. Algorithm $1$ carefully handles two cases: fewer clients than domains ($K < M$) and more clients than domains ($K\geq M$). In either case, each client holds non-overlapping domains, we term this case as **domain separation**. Then, we provide a convex combination between a uniform splitting of domains among clients (i.e., the $\frac{n_m}{K}$ term, where $n_m$ is the is the number of samples for domain $m$) and a domain separation splitting where each client has a disjoint set of domains. Depending on the convex parameter $\lambda,$ we can cover homogeneous case $\lambda=1,$ heterogeneous case $\lambda\in(0,1),$ and domain separation case $\lambda=1.$
>
> We would like to point out that the case that the Reviewer is mentioning in this question corresponds to  when each client only has access to one domain locally, which is a special case of domain separation, where $K\geq M.$
>
>
> **Reply to (ii)** The Reviewer is right, when encountering this case ($\lambda=0, K\geq M$), each client locally only has access to one domain, those centralized domain generalization methods need multiple domain information, thus they can not be directly applied to these case.
> In fact, we only compare ERM, FedDG, FedADG, FedGMA, FedSR in this case (note that we add FedSR and FedGMA in the revised version). The Reviewer may want to look at Section 4.1 for PACS and Section 4.2 for IWildCam and CivilComments in this scenario.
>
> We kindly ask the Reviewer to look at the revised version **Section 3.2 under Adapting Centralized DG Methods to FL Setting** for clarification of this point. To summary, to adapt centralized methods, we simply run the centralized DG method at each client locally with their own local dataset and then compute an average of model parameters at each communication round. This approach is straightforward for the homogeneous ($\lambda=1$) and heterogeneous ($\lambda=0.1$) settings where each client has data from all domains---albeit quite imbalanced for $\lambda=0.1,$
> where it can be seen as biased updates at each client based on biased local data. Similarly, this approach can be implemented in the domain separation case \emph{if} all clients hold \emph{multiple} non-overlapping domains (i.e., $\forall k, |P_k| \geq 2$, which could happen if $M\geq 2K$). Please check Section 4.1 for EMNIST and Section 4.2 Py150 for this domain separation case.

---

> ### Author Response · Authors · 2022-11-19
> **Author Response to Reviewer UVoT | Part 3 of 4**
>
> > "Although the performance is evaluated it is not discussed if these approaches come with additional communication or training time overhead"
>
> Thank you for the constructive comment. We highlight communications rounds and training epochs for each experiment in the start of Section 4.1 and Section 4.2, we would like to stress here that they are the same for all the methods to make a fair comparison in each experiment. We would also like to mention that in Section 4.3, we also considered different number of communication round $C$, note that we change the local training epochs $E$ when communication budget $C$ changes, that is, if the regime restrict the communication budget, then we increase its local computation $E$ to have the same the total computations. Therefore, the comparison on the influence of communication is fair  between algorithms  because the total data pass is fixed.
>
> However, we agree with the Reviewer that the training time should also be reported. Specifically, we report in Table 17 to elaborate the walk-clock training time for each method in Section C in the Appendix.
>
> Note that the training time for centralized DG methods are roughly the same for each dataset,  while FedDG and FedADG are extremely slow due to the following reasons: FedDG requires Fourier transformation during training, for and FedADG, each client locally trains a GAN model, resulting the training is slow. We also report the FedGMA suggested by the Reviewer and FedSR in Table 17.
>
> > "Related to the previous point are there additional information besides the models transmitted across clients in the various methods"
>
> About the privacy concern, we kindly ask the reviewer to look at the Table 5, where we listed the privacy constraint for each method. To summary, as we mentioned in the Section 2 background and related work, FedDG violates the privacy constraint in the FL context, because it requires sharing the amplitude spectrum of images among local clients. Except for FedDG, all the other methods, only transmit the models to the server, and they all respect the privacy constraint.

---

> ### Author Response · Authors · 2022-11-19
> **Author Response to Reviewer UVoT | Part 4 of 4**
>
> >``The datasets are interesting however in addition to these it might be good to include at least one popular dataset from the FL literature such as FEMNIST (which has been studied in OOD settings in other work) or CIFAR"
>
> Good point. We appreciate the Reviewer pointing the FEMNIST to study OOD generalization. We have added it in the revised version, the Reviewer may want to look at Section 4.1 under FEMNIST-62 (digits and characters). In particular, we report the performance of ERM, FISH, Coral, GroupDRO,  IRM, MMD, Mixup and FedSR, FedGMA for three different domain heterogeneous cases. Note that FedDG and FedADG are not implemented on FEMNIST yet give time limits, because FedADG need GAN trainning and FedDG needs Fourier transformation, we will add these two in the final version.
>
> Currents results on FEMNIST are summarized in Figure 3 for accuracy versus communication rounds in the main text, and the in-domain accuracy are reported in Table 7, Section C in the Appendix.
> The experiments on FEMNIST are meaningful in the following senses: 1) we use it as a demonstration for domain separation when $K<M.$ Thus, all of the methods are applicable on this dataset; 2) we observe that as the domain balance parameter $\lambda$ decreases from $1$ to $0,$ FedGMA, FedSR are consistently comparable to ERM while the others fail.
>
> We would like to mentioned that CIFAR dataset does not contains multiple domains, thus we did not include it in our papers to study domain generalization.
>
> We also added another real-world NLP dataset CivilComments, we evaluate IRM, FISH, MMD, Coral, Group DRO, FedSR, FedGMA on this dataset. Note Mixup, FedDG, and FedADG are only suitable for image datasets, thus are excluded in the experiments on CivilComments. The Reviewer may want to look at Table 3 for a test accuracy with held-out-domain validation, and Table 10 in the appendix for the test accuracy with in-domain validation. To observe the convergence of each algorithm, we also plot accuracy versus communication rounds in Figure 7, Section C.

---

> ### Author Response · Authors · 2022-11-30
> **Thanks for the comments!**
>
> We thank again the reviewer UVoT for the valuable comments which significantly help us strength our work, and thank him/her for acknowledging our work and raising the score.

---

### Author Response · Authors · 2022-11-19
**Response to All Reviewers**

We thank the reviewers for recognizing the importance (e.g., ``The DG problem under FL is important and lack of discussion, the proposed benchmark can better promote the research of new algorithms``,``The evaluations of this work are detailed covering a variety of methods``,``The work highlights the difficulties in this setting``, ``The list of algorithms included in the benchmark is also comprehensive,`` ``This is important work needed to advance the field,`` ``Domain generalization (DG) in federated learning (FL) is an important problem and has not been a systematic comparison of the methods. This paper presents the first such study``) and impact (e.g., ``The first paper to benchmark DG in FL settings and considered different setups and difficulties in the benchmark,`` ``Using homogeneous and heterogeneous splits (as measured by lambda) presents a systematic way to measure how well the algorithm can adapt to heterogeneous environments``, ``Sufficient experimental results and reproducible code``, ``The experimental settings for statistical heterogeneity challenge under FL is novel and practical``) of a federated DG benchmark.
Per the reviewers' comments, we have significantly updated the paper (changes in blue) summarized as follows:

1. We have added two new datasets FEMNIST (suggested by Reviewer UVoT) and CivilComments, an NLP dataset in addition to our text-based Py150 dataset.

2. We have added two new SOTA FL DG methods FedSR (accepted to the upcoming NeurIPS 2022) and FedGMA (suggested by Reviewer UVoT), both of which were released after the original submission.

3. We have completely rewritten the problem setup to formalize our setup and clarify important details. We  also added a *new figure* and a *new algorithm* to promote clarity (see Section 3).

4. We have done a significant rewrite of the experiments section for clarity with the new datasets and methods to expand upon our original submission (see Section 4).

More details are given in our responses to each reviewer.

---

> ### Author Response · Authors · 2022-11-29
> **A kind remind of discussion**
>
> We thank the reviewers once again for their positive reviews. We encourage the reviewers to reassess our paper in light of our responses and updates, and reach out with any other concerns and questions; we are happy to continue the discussion.

---

> ### Author Response · Authors · 2022-12-05
> **Remaining experiment results**
>
> We complete and report here the experiment results regarding the new datasets and methods suggested by the reviewers (which were left blank in the submitted manuscipts due to time limits).
>
> Table 4(a): FedSR on py150
> |           |centralized|$\lambda=1$|$\lambda=0.1$|$\lambda=0$
> |--------|------------|------------|------------|-------------|
> |FedSR|0.4626|0.0915|0.0860|0.0587|
>
> Table 7: Test accuracy on FEMNIST dataset with two model selection criterion: in-domain / held-out-domain validation; total client number K = 100; total training domain number M = 3500.
> |           |centralized|$\lambda=1$|$\lambda=0.1$|$\lambda=0$|centralized|$\lambda=1$|$\lambda=0.1$|$\lambda=0$|
> |--------|------------|------------|------------|-------------|------------|------------|------------|-------------|
> |FedDG|0.8582|0.8591|0.8573|0.8576|0.8592|0.8581|0.8567|0.8576|
> |FedADG|0.8488|0.0549|0.0557|0.0548|0.8594|0.0549|0.0557|0.0548|
>
> Table 8: Test accuracy on Py150 dataset with two model selection criterion: in-domain / held-out-
> domain validation; total client number K = 100.
> |           |centralized|$\lambda=1$|$\lambda=0.1$|$\lambda=0$|centralized|$\lambda=1$|$\lambda=0.1$|$\lambda=0$|
> |--------|------------|------------|------------|-------------|------------|------------|------------|-------------|
> |FedSR|0.4626|0.0915|0.0860|0.0587|0.4626|0.0915|0.0860|0.0587|
>
> Table 10: Test accuracy on IWildCam dataset with two model selection criterion: in-domain / held-
> out-domain validation; total client number K = 243.
> |           |centralized|$\lambda=1$|$\lambda=0.1$|$\lambda=0$|centralized|$\lambda=1$|$\lambda=0.1$|$\lambda=0$|
> |--------|------------|------------|------------|-------------|------------|------------|------------|-------------|
> |FedGMA|N.A.|0.0271|0.0259|0.0081|N.A.|0.0271|0.0259|0.0081|
>
> Table 11: Test accuracy on PACS dataset with two model selection criterion: in-domain / held-out-
> domain validation; total client number K = 20; total training domain number M = 2.
> |           |centralized|$\lambda=1$|$\lambda=0.1$|$\lambda=0$|centralized|$\lambda=1$|$\lambda=0.1$|$\lambda=0$|
> |--------|------------|------------|------------|-------------|------------|------------|------------|-------------|
> |FedGMA|N.A.|0.9413|0.9443|0.9599|N.A.|0.9377|0.9437|0.9521|
>
> Table 12: Test accuracy on PACS dataset with two model selection criterion: in-domain / held-out-
> domain validation; total client number K = 50; total training domain number M = 2.
> |           |centralized|$\lambda=1$|$\lambda=0.1$|$\lambda=0$|centralized|$\lambda=1$|$\lambda=0.1$|$\lambda=0$|
> |--------|------------|------------|------------|-------------|------------|------------|------------|-------------|
> |FedGMA|N.A.|0.7641|0.9251|0.9299|N.A.|0.8251|0.9425|0.9449|
>
> Table 13: Test accuracy on PACS dataset with two model selection criterion: in-domain / held-out-
> domain validation; total client number K = 100; total training domain number M = 2.
> |           |centralized|$\lambda=1$|$\lambda=0.1$|$\lambda=0$|centralized|$\lambda=1$|$\lambda=0.1$|$\lambda=0$|
> |--------|------------|------------|------------|-------------|------------|------------|------------|-------------|
> |FedGMA|N.A.|0.8820|0.8467|0.8180|N.A.|0.9078|0.8557|0.8892|
>
> Table 14: Test accuracy on PACS dataset with two model selection criterion: in-domain / held-out-
> domain validation; total client number K = 200; total training domain number M = 2.
> |           |centralized|$\lambda=1$|$\lambda=0.1$|$\lambda=0$|centralized|$\lambda=1$|$\lambda=0.1$|$\lambda=0$|
> |--------|------------|------------|------------|-------------|------------|------------|------------|-------------|
> |FedSR|0.8247|0.1275|0.1281|0.1192|0.8754|0.1168|0.1168|0.1168|
> |FedGMA|N.A.|0.9377|0.8581|0.8916|N.A.|0.9551|0.9431|0.9257|
>
> Table 15: Test accuracy on PACS dataset with two model selection criterion: in-domain / held-out-
> domain validation; total client number K = 200; total training domain number M = 2; Communica-
> tion Rounds C = 5.
> |           |centralized|$\lambda=1$|$\lambda=0.1$|$\lambda=0$|centralized|$\lambda=1$|$\lambda=0.1$|$\lambda=0$|
> |--------|------------|------------|------------|-------------|------------|------------|------------|-------------|
> |FedSR|0.8246|0.1222|0.1228|0.1186|0.8754|0.1222|0.1228|0.1186|
> |FedGMA|N.A.|0.8850|0.9162|0.9359|N.A.|0.8850|0.9162|0.9359|
>
> Table 16: Test accuracy on PACS dataset with two model selection criterion: in-domain / held-out-
> domain validation; total client number K = 200; total training domain number M = 2; Communica-
> tion Rounds C = 10.
> |           |centralized|$\lambda=1$|$\lambda=0.1$|$\lambda=0$|centralized|$\lambda=1$|$\lambda=0.1$|$\lambda=0$|
> |--------|------------|------------|------------|-------------|------------|------------|------------|-------------|
> |FedSR|0.8246|0.1186|0.1180|0.1162|0.8754|0.1186|0.1180|0.1162|
> |FedGMA|N.A.|0.9287|0.9341|0.9389|N.A.|0.9299|0.9383|0.9503|

---

### Decision · Program_Chairs · 2023-01-20

**Decision:**

Reject

**Justification For Why Not Higher Score:**

The authors provided a strong effort to improve the paper with respect to reviewers' comments. 3 reviewers out of the 4 were positives for a weak accept after the rebuttal, even though there is a comment shared by 2 reviewers about the fact that some representative methods are missing, and one still thinks that the paper is not ready in particular due to presentation issues.

During discussion, the following elements were raised:
-interesting topic and contribution.
-differences with classic Domain Generalization can be better discussed.
-some approaches related to meta-learning and multi-task learning in FL are not considered and could affect the conclusions of the study.

Based on these last remarks, I tend to propose reject. In addition, many modifications were made and the authors could not integrate the very last results in the document (they are provided as additional comments), but the contribution was very different with respect to the original submission and cannot be evaluated as a complete paper, which could be a bit unfair with respect to other submissions.


**Justification For Why Not Lower Score:**

N/A

**Metareview: Summary, Strengths And Weaknesses:**

This paper proposes a benchmark for Domain Generalization (DG) in Federated Learning (FL). The settings considered by the authors takes into account large client numbers, different levels of heterogeneity and number of communication rounds. The experimental evaluation is done on 5 datasets (included 2 added after rebuttal) and compare 4 FL strategies with 6 classic methods in DG that can be considered as centralized algorithms  and include. One important conclusion of the work is that there is an important of work to do to address the problem of DG in FL and that this work contributes to provide a benchmark potentially useful for the community.


Strengths:
-the problem of DG in FL is interesting.
-the benchmarking study can be useful the community.
-a large covering is proposed.
-the work illustrates some difficulties of the settings.


Weaknesses:
-the similarities and differences with classic domain generalization problems could be better discussed and analyzed.
-the study contains many approaches but some recent methods are not covered, in particular the number of FL methods considered is limited in some sense
-Some approach in FL related to meta-learning and multi-task learning are not considered in the study while potentially adapted to the context and could modify the analysis. More representative methods about DG and FL could be added.
-the number of changes made in the paper were numerous but the final version did not contain all the results that were finally provided as additional comments. A general review final submission is then hard to make.



Overall, the authors provided a strong effort to take into account the feedback of the reviewers and the majority of the reviewers were positively impressed by the work done.
However, the number of FL methods considered is somewhat limited and in particular ones related to meta-learning and multi-task learning which are relevant for the problem and can be important to draw a general conclusion. The differences with general DG can be then more discussed. Since these elements are important for a benchmark paper that can be impactful for the community, I think that the contribution is still not ready and I propose rejection.


Example of Meta-learning and multi-task learning in FL that can be considered:
*Alireza Fallah, Aryan Mokhtari, Asuman E. Ozdaglar: Personalized Federated Learning with Theoretical Guarantees: A Model-Agnostic Meta-Learning Approach. NeurIPS 2020.
https://arxiv.org/abs/2002.07948.
*Othmane Marfoq, Giovanni Neglia, Aurélien Bellet, Laetitia Kameni, Richard Vidal: Federated Multi-Task Learning under a Mixture of Distributions. NeurIPS 2021.
https://arxiv.org/abs/2108.10252.


Other representative DG methods:
*Jigen - "Domain generalization by solving jigsaw puzzles." CVPR 2019.
*RSC - "Self-challenging improves cross-domain generalization" ECCV 2020.
*FACT - "A Fourier-based framework for domain generalization" CVPR 2021.
*SWAD - "Swad: Domain generalization by seeking flat minima." NeurIPS 2021.

Other representative FL methods:
*Scaffold - "Scaffold: Stochastic controlled averaging for federated learning" ICML 2020.
*FedDyn - "Federated learning based on dynamic regularization" ICLR 2021.
*HarmoFL - "Harmofl: Harmonizing local and global drifts in federated learning on heterogeneous medical images."AAAI 2022.
*FedSAM - "Generalized federated learning via sharpness aware minimization." ICML 2022.
*FedAlign - "Local Learning Matters: Rethinking Data Heterogeneity in Federated Learning" CVPR 2022.